# Near surface structure of the North Anatolian Fault Zone from Rayleigh and Love wave tomography using ambient seismic noise.

George Taylor[1,3], Sebastian Rost[1], Gregory Houseman[1], and Gregor Hillers[2,3]

[1]School of Earth and Environment, University of Leeds, UK, LS2 9JT
[2]Institut des Sciences de la Terre, Université Grenoble-Alpes, France, F-38041
[3]Now at Institute of Seismology, University of Helsinki, Finland, 00014

**Correspondence:** George Taylor (george.taylor@helsinki.fi)

**Abstract.** We use observations of surface waves in the ambient noise field recorded at a dense seismic array to image the North Anatolian Fault Zone (NAFZ) in the region of the 1999 magnitude 7.6 Izmit earthquake in western Turkey. The NAFZ is a major strike slip fault system extending $\sim 1200$ km across northern Turkey and poses a high level of seismic hazard, particularly to the city of Istanbul. We obtain maps of phase velocity variation using surface wave tomography applied to Rayleigh and Love waves and construct high resolution images of S-wave velocity in the upper 10 km of a 70 km by 30 km region around Lake Sapanca. We observe low S-wave velocities ($< 2.5$ km s$^{-1}$) associated with the Adapazari and Pamukova sedimentary basins, as well as the northern branch of the NAFZ. In the Armutlu Block, between the two major branches of the NAFZ, we image higher velocities ($> 3.2$ km s$^{-1}$) associated with a shallow crystalline basement. We measure azimuthal anisotropy in our phase velocity observations, with the fast direction seeming to align with the the strike of the fault at periods shorter than 4 s. At longer periods, up to 10 s, the fast direction aligns with the direction of maximum extension for the region ($\sim 45°$). The signatures of both the northern and southern branches of the NAFZ are clearly associated with strong gradients in seismic velocity that also denote the boundaries of major tectonic units. Our results support the conclusion that the development of the NAFZ has exploited this pre-existing contrast in physical properties.

## 1 Introduction

The formation of fault zones appears to be a balance between the accommodation of the tectonic strain field, and the exploitation of pre-existing weak zones such as tectonic suture zones or lithological boundaries (e.g. Bercovici and Ricard (2014), Dayem et al. (2009), Gerbi et al. (2016), Tapponier et al. (1982)). Studying how structural changes affect strain localisation in the upper crust is critical to understanding the earthquake cycle (Bürgmann and Dresen, 2008). Imaging the seismic velocity structure of fault zones provides information essential to understanding the long-term behaviour of faults and the earthquakes that occur on them.

Here we interpret images from ambient noise surface wave tomography of the upper 10 km of the North Anatolian Fault Zone (NAFZ), Turkey, in the rupture zone of the 1999 Izmit earthquake. This allows us to study the near surface structure of a recently ruptured fault. The NAFZ is a $\sim 1200$ km long strike slip fault that forms the boundary between the Anatolian block and the Eurasian continent. Progressively localized since the middle Miocene ($\sim 3$ Ma), the NAFZ propagated westward from the Karliova Triple Junction in eastern Turkey, across northern Anatolia, and reached the Izmit-Adapazari region $\sim 200$ ka, although a more broad zone of shear deformation was present since the middle Miocene (Sengör et al., 2005). The motion of Anatolia is driven by a gradient of lithospheric gravitational potential energy that extends across the Anatolian Peninsula (England et al., 2016) and is sustained by the collision between the Arabian and Eurasian plate in the East, and the roll-back of the Hellenic trench to the southwest (Flerit et al., 2004; Reilinger et al., 1997). Since 1939 a westward propagating sequence of large earthquakes ($M_w > 7.0$) has occurred along the NAFZ (Stein et al., 1997). The 1999 Izmit ($M_w$ 7.6) and Düzce ($M_w$ 7.2) earthquakes are the most recent in this sequence (Barka et al., 2002), and the NAFZ continues to pose significant seismic hazard to the region.

In the Izmit-Adapazari region, the NAFZ is split into northern and southern branches (Fig. 1). The northern branch has seen more seismic activity historically, but micro-seismicity in this region does not appear to be strongly localised to the major fault strands (Altuncu Poyraz et al., 2015). The northern branch of the fault appears to exploit the so-called Intra-Pontide Suture between the Eurasian continent and sedimentary accretionary complexes formed during the closure of the Tethys Ocean (Okay, 2008). There are three major geological units delineated by the fault zone (Fig. 1). To the north of the northern branch of NAFZ is the Istanbul Zone, a cratonic fragment of the Eurasian continent. The Istanbul Zone includes the Adapazari Basin, a $\sim 2$ km thick pull-apart sedimentary basin formed by right-lateral motion acting on a change in strike of the northern branch of the NAFZ (Sengör et al., 2005).

Located between the two fault branches are the Armutlu Block and the Almacik Mountains. The Armutlu Block is a section of the Almacik mountains that has migrated further westward with motion along the NAFZ. Both are areas of high topography, formed as an accretionary complex of upper cretaceous sediments overlying a metamorphic basement (Yılmaz et al., 1995). The dominant feature of the Armutlu Block is an abundance of metamorphosed sediments and marbles of unknown age and provenance (Okay and Tüysüz, 1999). The Pamukova sedimentary basin is located in the southern part of the Armutlu Block (Fig. 1). Striations and down dip motion on faults observed along the southern branch of the NAFZ in the Pamukova basin (Doğan et al., 2014) indicate that extension in the NE - SW direction due to the right lateral motion is more dominant than shortening in the NW - SE. The resulting transtensional strain is believed to have caused the opening of the Pamukova basin (Doğan et al., 2014). The total thickness of the sediments in the Pamukova basin is generally unknown, but it is thought to be thinner than in the Adapazari basin (Sengör et al., 2005).

To the south of the NAFZ lies the Sakarya Terrane, an accretionary complex of sedimentary rocks from the Jurassic - lower Cretaceous, overlying a metamorphic basement of mainly Paleozoic rocks (Yılmaz et al., 1995). The Sakarya Terrane also contains a number of ophiolitic melanges, including serpentinites close to the southern branch of the NAFZ that were probably produced by imbrication and thrust-stacking during the closure of the Neo-Tethys Ocean (Sengör and Yılmaz, 1981).

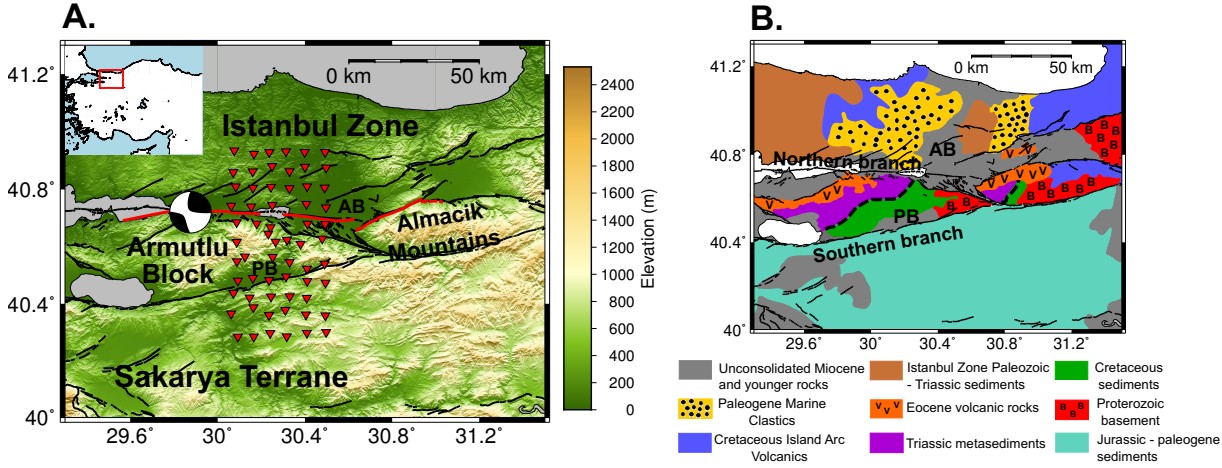

**Figure 1.** A.: Overview of the Izmit-Adapazari region and the DANA network. Stations of the DANA network are shown as red triangles; station names are of the form Dx01 to Dx11, where x is A through F from west to east and 01 is at the southern end of each line. Thick black lines identify mapped faults in the region (Emre et al., 2016). The thick red line indicates the extent of the rupture of the 1999 Izmit and Düzce earthquakes (Barka et al., 2002). The epicenter and focal mechanism for the Izmit earthquake provided by the GCMT catalogue (Dziewonski et al., 1981; Ekström et al., 2012) is shown. Topography data were acquired by the Shuttle Radar Topography Mission (USGS, 2006). B.: Geological map of the Izmit-Adapazari region, simpified from Akbayram et al. (2016). The location of the southern and northern branches of the North Anatolian Fault Zone are indicated. The black dashed line shows the location of the Intra-Pontide Suture within the Armutlu Block inferred by Akbayram et al. (2016). AB and PB show the location of the Adapazari and Pamukova Basin, respectively.

To study the structure of the NAFZ in the Izmit-Adapazari region the University of Leeds, Kandilli Observatory and Earthquake Research Institute (KOERI) and Sakarya University deployed a temporary array of seismometers across the rupture zone of the 1999 Izmit earthquake between May 2012 and October 2013 (Kahraman et al., 2015). The array included 62 three-component seismometers in a 70 km x 35 km rectangular grid (Fig. 1), and an approximate station spacing of 7 km, known

as the Dense Array for Northern Anatolia (DANA, 2012). Also included were three stations of the KOERI national network located within the main grid of the DANA array: GULT, SAUV and SPNC. DANA was deployed over both strands of the NAFZ in this region, with stations sited on all three of the major crustal units described above (Fig. 1).

Short period surface waves from ambient noise have been used to study the upper crust in the vicinity of active fault zones in the past (e.g. Lin et al. (2013), Zigone et al. (2015)). In such studies low seismic velocities have been attributed to earthquake

damage zones and pull-apart sedimentary basins. Here our analysis of the DANA data provides an image of the top 10 km of the NAFZ in the Izmit-Adapazari region with a lateral resolution dictated by the $\sim 7$ km station spacing, to better constrain the relationship between the fault and its regional geological context.

We also interpret first order observations of azimuthal anisotropy within our phase velocity measurements. Observations of azimuthal anisotropy in the upper crust can provide insights into the state of tectonic stress within a region, and potentially the

orientation of pervasive mineral fabric and the structural influence of major faults (e.g. Hurd and Bohnhoff (2012), Polat et al.

(2012)). Such information provided by azimuthal anisotropy is particularly important in areas such as the North Anatolian Fault, where in-situ stress observations are rare, and extensive deformation occurs off of mapped faults (Bouchon and Karabulut, 2008; Altuncu Poyraz et al., 2015). Earthquake focal mechanisms suggest that the direction of maximum compressive stress in the Izmit-Adapazari region is oriented NW – SE, between 120° – 160° from north (Bohnhoff et al., 2006). If the regional anisotropy is primarily stress-controlled, we would expect the seismic fast direction to be aligned in the direction of maximum compressive stress, due to the preferential closure of fractures in this direction (Crampin and Lovell, 1991). However, Peng and Ben-Zion (2004) used local seismicity to show that the fast polarisation direction at stations close to the ruptured Düzce fault (Fig. 1) are generally parallel to and vary with the fault strike, suggesting an anisotropy mechanism determined by deformation fabric. They suggested that the anisotropy is confined to the top 3 – 4 km of the crust.

Using local seismicity recordings from other stations in the Izmit-Adapazari region, more distant from the ruptured fault, Hurd and Bohnhoff (2012) found a more complex pattern, with the fast polarisation directions for at least three of their stations consistent with the maximum compressive stress direction (approximately NW – SE). They concluded that anisotropy is limited to depths less than 8 km. Further east, on the central section of the North Anatolian Fault system, Biryol et al. (2010) used teleseismic data to find a coherent anisotropy signature attributed to mineral fabric within the mantle lithosphere, in which the fast polarisation direction aligns with the principal extension direction (approximately NE – SW). These results indicate that stress orientation controls shear wave anisotropy in places, but mineral fabric dominates in others. By providing further analysis of the regional anisotropy through surface wave phase velocities, we expect to provide more observations that can contribute to better understanding of the various mechanisms that cause seismic anisotropy in the upper crust.

## 2    Data and Methods

### 2.1    Calculation of the cross-correlation functions

To image the upper 10 km of the NAFZ we used ambient noise data recorded at DANA to construct cross-correlation functions and retrieve empirical estimates of the elastic Green's function of the Earth for all inter-station paths of the network (Lobkis and Weaver, 2001; Campillo and Paul, 2003; Shapiro and Campillo, 2004; Wapenaar, 2004). The instruments used for the DANA network were all three-component broadband sensors, the majority of which were Guralp CMG-6TDs (30 s maximum period). Some stations were equipped with CMG-3Ts or CMG-3ESPs (120 s maximum period). From these cross-correlation functions we extract surface wave dispersion curves in order to perform seismic tomography and invert for S-wave velocity structure (Shapiro et al., 2005). The data were first reduced to a 25 Hz sampling rate and corrected for the instrument response. An initial band-pass filter was applied between 0.02 and 10 Hz, and the frequency spectrum of each noise window was whitened between 0.05 and 2 Hz (Bensen et al., 2007). We tested several pre-processing methods for producing the cross-correlation functions for this study. These included trialling the use of 4-hour and 1-hour long noise windows. In order to remove any data windows containing signals from large earthquakes, each window was split into three segments. If the amplitude of one of these segments has a significantly higher standard deviation ($> 1.8$ times) than the other two, the data window is discarded (Poli et al., 2012). For amplitude normalisation (Bensen et al., 2007), we tested 1-bit normalisation against clipping any data with an

amplitude > 3.5 times the standard deviation of each data window. Supplementary Figs. S1 and S2 show the results of these tests. We found little difference between the processing schemes in terms of signal-to-noise ratio of the final cross-correlation functions. However, the approach of amplitude clipping for 4-hour long noise windows was found to produce correlation functions with slightly higher frequency domain coherence than the other schemes. As such, we selected this pre-processing method.

Following this pre-processing, each data window is cross correlated with the corresponding window at every other station in the network, and these cross correlations are then stacked over the entire duration of the array deployment (16 months of data). We calculated the correlations for all 9 possible combinations of the vertical, north and east components of ground motion, and then rotated the final stacked correlations into the relevant great circle path (station to station) to retrieve the vertical, radial and transverse correlation components (Fig. 2). The correlation functions in Fig. 2 are stacked in bins of 0.5 km interstation distance, and band-pass filtered between 0.05 and 2.0 Hz. The amplitudes are normalised within each bin.

## 2.2  Extraction of surface wave phase velocities

The record sections exhibit multiple features and arrivals. There are two explanations for the large-amplitude features around t = 0. Firstly, they may represent the signature of the overlapping converging and diverging surface waves to form focal spots in the wave field (Hillers et al., 2016). A second possible explanation is teleseismic body wave energy that arrives at the stations at a near-vertical incidence angle. When these arrivals are cross-correlated, the very small differential travel times of the energy result in large amplitudes near the zero lag correlation time (Landès et al., 2010; Hillers et al., 2013). The large amplitudes are particularly prominent on the ZZ component. This phenomenon has been observed in a previous ambient noise study in Turkey: Warren et al. (2013) observe large zero-time amplitudes in their correlation functions up to a distance of 80 km. Additionally, large amplitude waveforms near t = 0 are often observed in ambient noise correlation studies (e.g. Poli et al. (2012), Villaseñor et al. (2007), Zheng et al. (2011)), and are typically left uninterpreted. While these waveforms can be used for imaging (e.g. Hillers et al. (2016), Taylor et al. (2016)), we focus here on the propagating surface waves that dominate the record sections. Correlations between the vertical and radial components (ZZ, ZR, RR and RZ) predominantly contain Rayleigh waves propagating between DANA stations, whilst the transverse (TT) correlations contain Love waves. Fig. 2 shows some evidence for cross talk between vertical and transverse components (ZT and TZ) in the form of low amplitude coherent waves, perhaps indicating the effects of anisotropy or the scattering of waves off 3D earth structure. Linear arrivals that are most prominent at arrival times of ± 10 s may represent body wave reflections contained within the ambient noise, but may also be an artefact produced by the GPS time synchronisation of the seismic instruments (Lehujeur et al., 2018).

To create phase velocity dispersion curves for the study region, we first create group velocity - period diagrams (Levshin and Ritzwoller, 2001) for each stacked correlation function between periods of 1.0 s and 10.0 s (supplementary Fig. S3) using the program *do_mft* (Herrmann, 2013). We then pick the dispersion curve for each correlation function manually. Due to a poorer signal to noise ratio on the ZZ component, Rayleigh wave dispersion measurements are picked from the RR component correlations, whilst Love wave measurements are picked from the TT component. Examples of period - group velocity maps used for picking the dispersion curves are shown in supplementary Fig. S3. Bensen et al. (2007) suggest that in order for

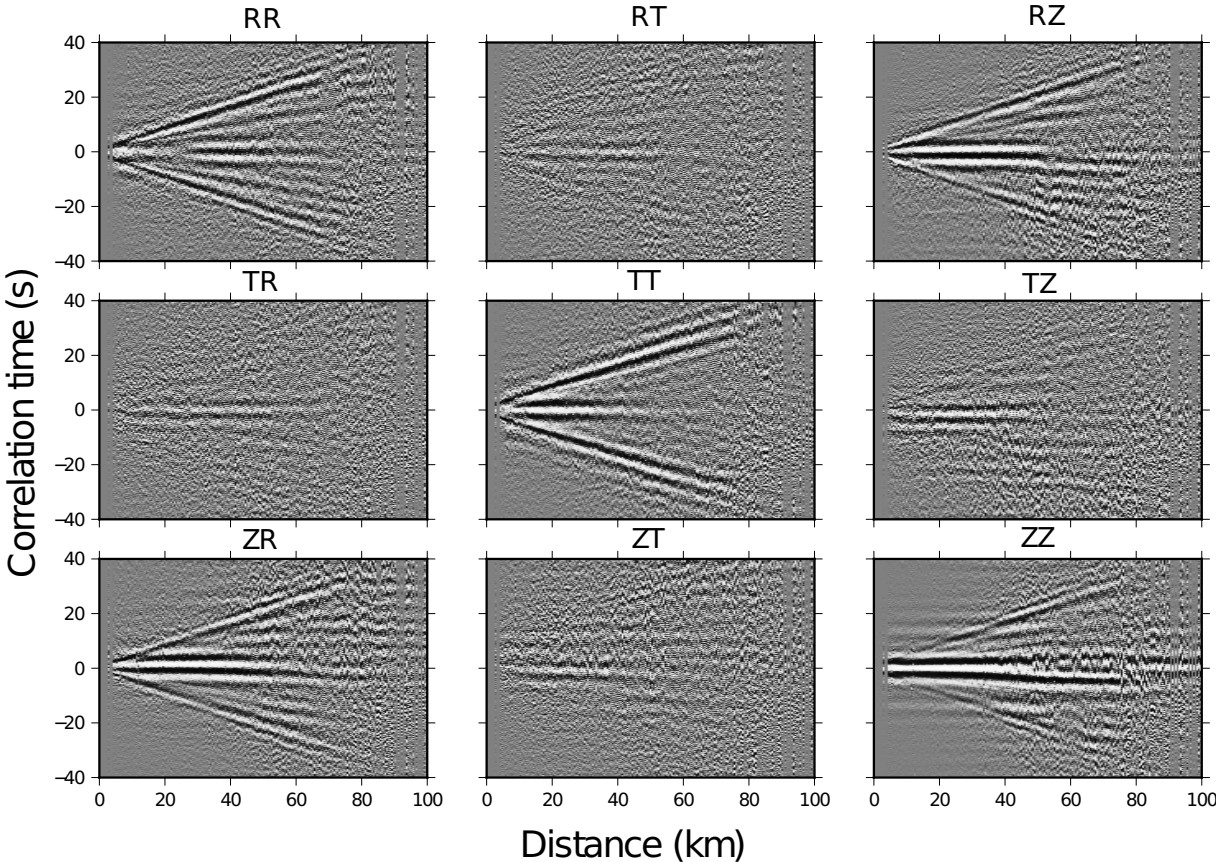

**Figure 2.** Record section of correlation functions calculated for inter-station paths of the DANA network. Correlation functions were filtered between 0.05 and 2.0 Hz and binned and stacked in 0.5 km distance bins, and the amplitude is normalized within each bin. Record sections for every combination of three component motion are labelled as follows: Z = vertical, R = radial, T = transverse. E.g. The ZR correlation (bottom left) represents the motion recorded on the radial component due to a vertical point source. ZZ, ZR, RR, RZ components show Rayleigh waves, and TT shows Love waves.

dispersion measurements to be considered reliable, the station separation must be greater than 3 wavelengths of the target wave. If we assume an average phase velocity of c = 3 km s$^{-1}$ for the upper crust, our shortest period surface waves of 1.5 s will have a wavelength of 4.5 km. Thus, in order to satisfy the wavelength criterion, we discard all measurements with an inter-station distance of 13.5 km or less as unreliable. For longer periods and inter-station distances, where some of the short period data may be trustworthy, unreliable long period measurements are discarded based on visual inspection. This also ensures that the large amplitudes of the near-zero arrivals do not contaminate our measurements from the later arriving surface waves. We use 62 stations in this study, which amounts to a total of 1891 unique station pairs. As a result of the wavelength criterion, coupled with the visual inspection of each period - velocity map we retain measurements from 929 station pairs for Rayleigh waves (49% of the RR correlations), and 1173 station pairs for Love waves (62% of the TT correlations).

Phase velocity dispersion curves are also picked using *do_mft* (Herrmann, 2013). The phase velocity at each period is calculated from the previously picked group velocity by:

$$c = \frac{\omega_0 r}{-\Phi + \frac{\pi}{4} + \frac{\omega_0 r}{U_0} + N2\pi}, \tag{1}$$

where $\Phi$ is the instantaneous phase of a narrow bandpass filtered surface wave, $\omega_0$ is the centre frequency of the bandpass filter, $r$ is the inter-station distance, $U_0$ is the group velocity, and $N$ is some integer. The $N2\pi$ term in Eq. 1 introduces an ambiguity in the calculation of the phase velocity. To overcome this ambiguity, *do_mft* (Herrmann, 2013) uses Eq. 1 to generate a suite of dispersion curves corresponding to different values of $N$. To pick the correct phase velocities, we calculate the theoretical dispersion curve using an *a priori* seismic velocity model of the region (Karahan et al., 2001), and manually pick the calculated dispersion curve (Eq. 1) that most closely corresponds to the theoretical dispersion curve.

### 2.3 Phase velocity tomography

After picking phase velocity dispersion curves for all inter-station pairs for both Rayleigh and Love waves, we convert the phase velocity at each period into a travel time between the stations. We then use these travel time observations to invert for phase velocity as a function of position at each discrete period. We discretize each model as a 2D grid of phase velocity nodes. The phase velocity tomography is carried out in a spherical co-ordinate system (Rawlinson and Sambridge, 2005), with the node spacing (6.6 km in latitude and 7.6 km in longitude) comparable to the average horizontal separation of the stations of the DANA network. We begin each inversion with a constant velocity model, with the velocity set to the average observed phase velocity at the given period. We then invert the travel times for periods between 1.5 s and 10.0 s using the method of Rawlinson and Sambridge (2005). This is an iterative inversion, with each step consisting of calculating travel times through the current phase velocity model by wave front tracking using the Fast Marching Method (Sethian and Popovici, 1999). The inversion then seeks to minimise the objective function:

$$| \mathbf{g}(\mathbf{m}) - \mathbf{d_{obs}} |^2 + \epsilon \left( (\mathbf{m} - \mathbf{m_0})^T (\mathbf{m} - \mathbf{m_0}) \right), \tag{2}$$

where $\mathbf{g}(\mathbf{m})$ are the travel times through the current model, $\mathbf{d_{obs}}$ are the observed travel times from our dispersion data, $\epsilon$ is a variable damping factor, $\mathbf{m}$ and $\mathbf{m_0}$ represent the current model and the starting model respectively. The variable damping term is included in order to minimise unconstrained model parameters (phase velocities) by preventing them from straying too far from our initial constant velocity model. The choice of damping parameter, $\epsilon$, is somewhat subjective. It should be selected with the aim of achieving a balance between the variance of the perturbations in the final phase velocity model with respect to the initial model (a high variance indicates unrealistic values for unconstrained model parameters), and obtaining a satisfactory misfit to the observed travel time data. We constructed trade-off curves (supplementary Fig. S4) of final model perturbation variance vs. final data misfit for both the Rayleigh and Love wave inversions. We selected a damping factor of 40 $s^4$ $km^{-2}$ for Rayleigh waves as it provided a 68% reduction in the perturbation variance of the final model parameters (0.025 $(km\,s^{-1})^2$ to 0.008 $(km\,s^{-1})^2$) for only a 2% increase in data misfit (795 ms to 815 ms) at 4 s period. Likewise, for Love waves we choose a damping parameter of 60 $s^4$ $km^{-2}$ which provides a 75% reduction in final model variance (0.055 $(km\,s^{-1})^2$ to

0.014 (km s$^{-1}$)$^2$) for an 8% increase in misfit (670 ms to 730 ms). Increasing the damping parameter above these values leads to an increase in misfit to the observed data which we find unacceptable. These constant damping factors are applied to the inversions at every period (Figs. 3 and 4).

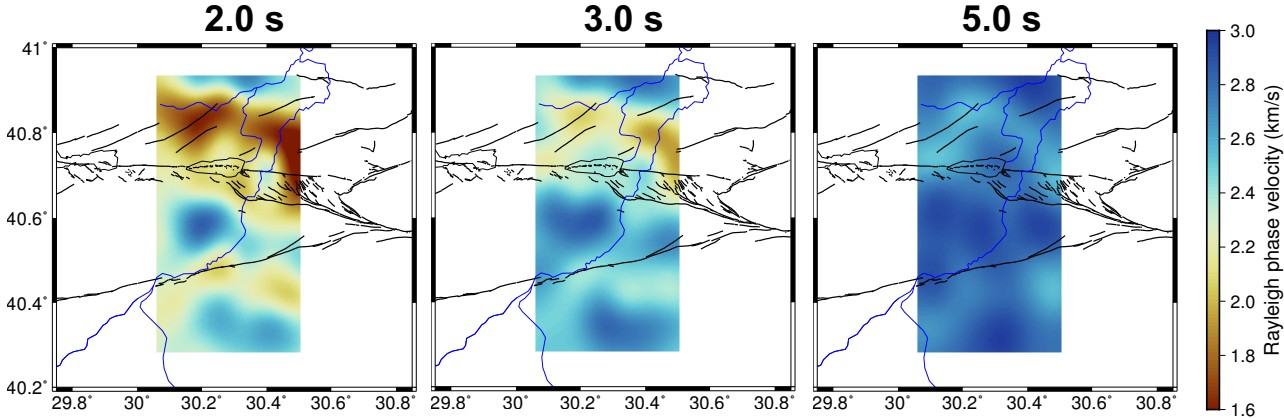

**Figure 3.** Rayleigh wave phase velocity maps at 2.0, 3.0 and 5.0 s period. Black lines show the mapped faults. The blue line represents the Sakarya River, flowing towards the north.

We do not include a separate smoothing parameter in our inversion scheme, as a similar effect can be obtained by simply reducing the number of model parameters and controlling the inversion through a damping parameter as described above (Rawlinson and Sambridge, 2003). We have designed our model discretization so that our velocity node separation is comparable to our station separation, which should be a sufficiently coarse parameterization to constrain all our model parameters, and produce a smooth final model.

The minimization of the objective function is performed using an iterative subspace inversion approach (Kennett et al., 1988), which projects the objective function on to a multi-dimensional subspace of the data and model parameters. After 10 iterations the data misfit does not improve appreciably with further iterations, and the inversion is judged to have converged. Stable solutions are shown in Figs. 3 and 4 for periods of 2.0, 3.0, and 5.0 s.

## 2.4 S-wave velocity inversion

After obtaining 2D maps of phase velocity for all periods between 1.5 and 10.0 s, the resulting dispersion relation at each node on the same geographic grid was inverted to obtain isotropic S-wave velocity as a function of depth at that location. Both Rayleigh and Love wave dispersion data are inverted together, with equal weighting, in order to obtain an S-wave velocity model that best satisfies both data sets. The initial inversion was performed using a neighbourhood algorithm (Sambridge, 1999b; Wathelet, 2008) parameterised by a model consisting of 10 layers with variable layer thickness and S-wave velocity. The total number of free parameters is 20. The S-wave velocity of each layer is permitted to vary with a uniform distribution between 0.5 and 4.5 km s$^{-1}$, whilst layer thickness could vary between 0.5 and 1.5 km. An increase of S-wave velocity with layer depth is also prescribed. The neighbourhood algorithm was allowed to run until 20050 different S-wave velocity models

had been generated for each node in the grid. The misfit parameter at each location is defined for the neighbourhood algorithm as:

$$\phi_m = \sqrt{\sum_{i=1}^{n_f} \frac{(v_{di} - v_{mi})^2}{v_{di}^2 n_f}}, \tag{3}$$

$n_f$ is the number of frequencies in the dispersion curve, $v_{di}$ is the observed phase velocity at frequency $i$ from our tomographic model, and $v_{mi}$ is the phase velocity at that frequency inferred from the inverted S-wave model. Models that fit the dispersion curves extracted from the phase velocity tomography with $\phi_m < 0.25$ (eq. 3) were used in a weighted average to construct an initial estimate for S-wave velocity vs. depth. Examples of the distribution of models used in the weighted average at three grid points, one each in the Sakarya Terrane, Armutlu Block and Istanbul Zone, are shown in Fig. 5. The weighting of each model is the inverse of its misfit to the dispersion data as described in eq. 3.

This average model was then used as the starting model for a linearised iterative inversion scheme as implemented in *surf96* (Herrmann, 2013). The inversion was judged to have converged once the root mean square change in the S-wave velocity model between iterations was negligible ($< 0.1$ km/s), usually within 6 iterations. The set of 1D models obtained from the linearised inversion represent our 3D S-wave velocity model for the region.

The advantage of the neighbourhood algorithm is that it provides a much broader overview of the acceptable parameter space for our S-wave velocity model, rather than inverting for a single model that best fits the data. The output of the neighbourhood algorithm (Fig. 5) also allows for an intuitive, if qualitative, understanding of potential uncertainty in our final S-wave velocity model. A disadvantage of the neighbourhood algorithm is that only a relatively small amount of model parameters can be included in the inversion ($\sim 30$), before the parameter space becomes too large to search efficiently (Sambridge, 1999a). This means that the neighbourhood algorithm can only constrain relatively simple models. For these reasons, we present the results of the neighbourhood algorithm (Fig. 5), but also perform a linearised inversion (Herrmann, 2013) to obtain a final model that better fits the data overall. This approach has been used previously in fault zone imaging (Hillers and Campillo, 2018), and attempts to strike a balance between presenting a model that satisfies the data, as well as giving a broader overview of the acceptable model space that is not available when using only a linearised inversion scheme.

## 3   Results

In this section, we describe the phase velocity maps derived separately for Rayleigh and Love wave travel time data. Sensitivity kernels representing the vertical resolution for Rayleigh and Love waves within our period range can be found in the supplementary material (Fig. S8), along with synthetic checker board recovery tests to illustrate the horizontal resolution of the inversion (Fig. S9 and S10). The initial and final data misfit of the tomography models for both Rayleigh and Love wave phase velocities are shown in supplementary Figs. S5 and S6. The significant reduction in the variance of the travel time residuals in the final models, on average about 50%, indicates that the final models better account for structural heterogeneity. Similarly, the higher variance of the final travel time residuals at shorter periods indicates stronger heterogeneity at shallow depths, or noisier phase velocity measurements at these periods.

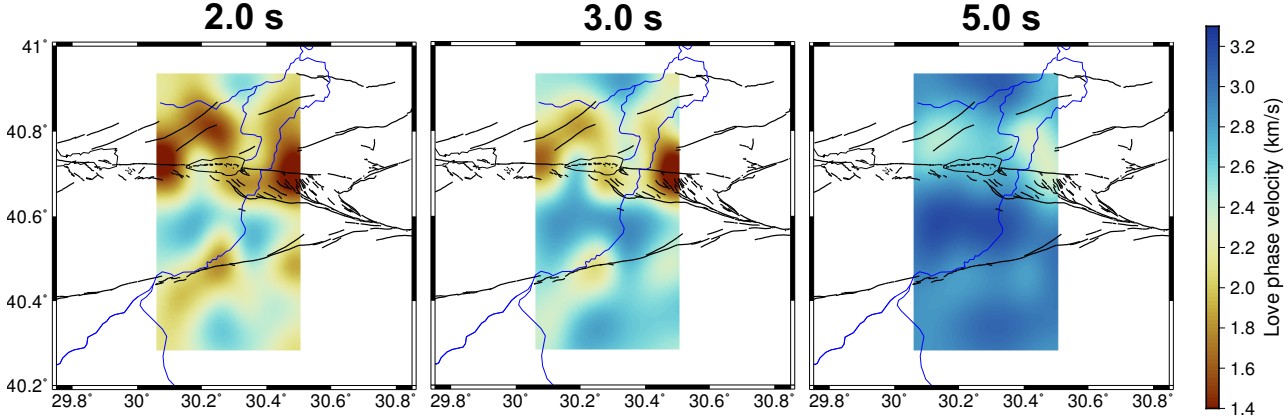

**Figure 4.** Love wave phase velocity maps at 2.0, 3.0 and 5.0 s period. Black lines show the mapped faults. The blue line represents the Sakarya River, flowing towards the north.

### 3.1 Rayleigh wave phase velocity

Fig. 3 shows the results of the Rayleigh wave phase velocity tomography for periods between 2.0 s and 5.0 s. The most interesting features of the velocity model include the large low velocity (1.5 km s$^{-1}$ - 2.0 km s$^{-1}$) anomaly located north of the northern branch of the NAFZ. These low velocities are likely due to the deep sedimentary basin at Adapazari in the north

eastern part of the model, and heavily faulted sediments near Izmit in the north western sector (Sengör et al., 2005). In between the two fault strands, the Armutlu Block can be seen as a prominent region of high phase velocity ($\sim$ 3.0 km s$^{-1}$), likely associated with the metamorphic rocks and possible granitic intrusions that exist in this region (Bekler and Gurbuz, 2008; Sengör et al., 2005). At 2.0 s and 3.0 s period, this high velocity region is particularly prominent in the western part of the Armutlu Block (Fig. 3). At 5.0 s period, the entire Armutlu Block consists of high velocities. At 2.0 s period, the sediments

of the Pamukova basin can be seen along the southern branch of the NAFZ with velocities of approximately 2.0 km s$^{-1}$. To the south, in the Sakarya Terrane, a relatively high velocity anomaly (faster than 2.5 km s$^{-1}$) can be seen at all periods greater than 2.0 s. These velocities are in general higher than those observed in the part of the Istanbul Zone that bounds the fault, and they likely indicate the crystalline basement of the Sakarya Terrane at shallower depths, with thinner sedimentary cover. It is likely that the high phase velocities observed in the far north of the model correspond to the older sedimentary units and

crystalline rocks of the Istanbul Zone that underlie the clastic sediments at Izmit and Adapazari (Okay et al., 1994). In general, at 5.0 s period and lower, the contrast in phase velocity between the major tectonic units is relatively low. This is likely due to the longer wavelength of these waves, which will average lateral variations in structure at these larger periods.

### 3.2 Love wave phase velocity

The Love wave phase velocity images (Fig. 4) show a very similar pattern to the Rayleigh wave images. To the north of

the fault extremely low ($\sim$ 1.2 km s$^{-1}$) phase velocities are associated with the faulted sediments near Izmit, as well as the

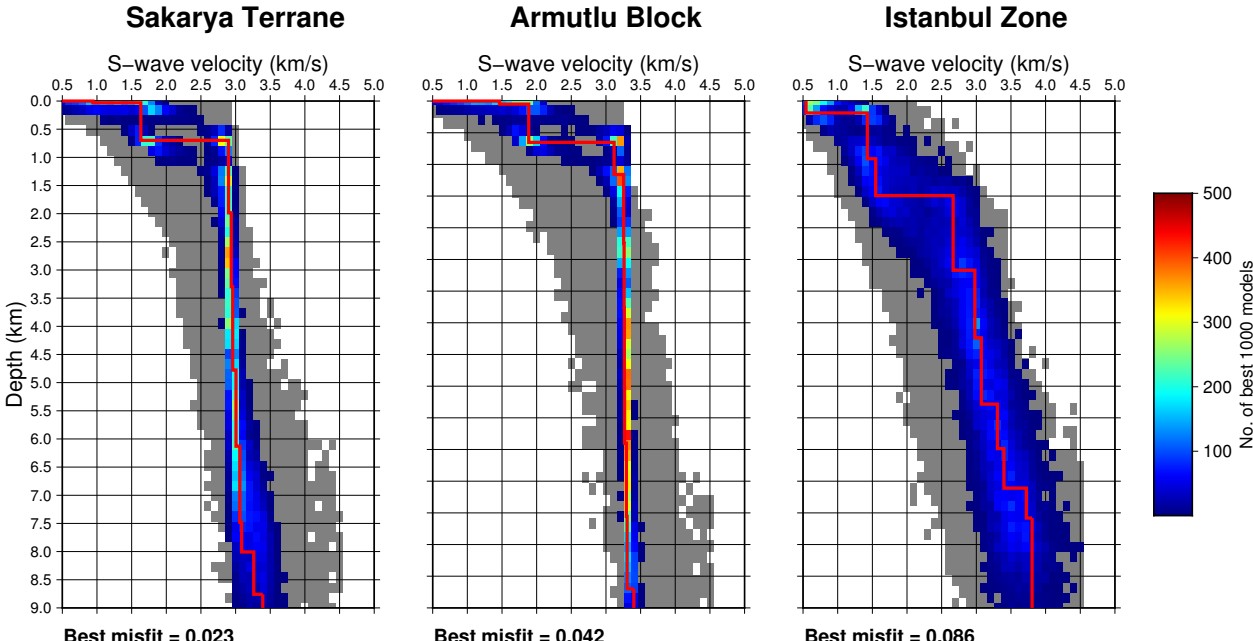

**Figure 5.** Results of the neighbourhood algorithm inversion for S-wave velocity at three nodes in the different geological units (Fig. 1). The grey region represents the range of accepted models with a misfit below 0.25 (eq. 3). The coloured region shows the range of the 1000 models with the lowest misfit. Red colours indicate a higher number of the best 1000 models with a certain S-wave velocity at that depth. The solid red line shows the best fitting model, the misfit of which is shown at the bottom of each panel. The location of each of these nodes is shown in Fig. 6.

Adapazari Basin. Both of these features are visible for periods < 5.0 s. Low velocities also seem to be strongly associated with the NW-SE striking faults just north of the rupture zone of the Izmit earthquake at 40.7°N and 30.45°E. Focal mechanisms for earthquakes in this region show examples of normal faulting (Altuncu Poyraz et al., 2015), indicating these low velocities could be associated with a releasing bend on the northern branch. The Armutlu Block in between the two fault strands shows

5   high phase velocities exceeding 2.4 km s$^{-1}$, which is comparable with those of the Rayleigh wave images. The Pamukova basin can be seen for periods < 5.0 s near the southern branch of the fault with velocities of 1.5 - 2.5 km s$^{-1}$. At 5.0 s period, higher phase velocities (> 3.0 km s$^{-1}$) are observed within the southern portion of the Sakarya Terrane, and the northern part of the Istanbul Zone. These high velocities are again interpreted to represent the crystalline basement of these tectonic units. As with the Rayleigh wave phase velocity maps (Fig. 3), the lateral resolution of the Love wave images decreases with increasing

10   period.

### 3.3   S-wave velocity model misfit

In order to construct an isotropic S-wave velocity velocity profile at each node a two-step inversion process was chosen, as described in sec. 2.4. Examples of the results of the neighbourhood algorithm from three locations in the Sakarya Terrane,

Armutlu Block and Istanbul Zone, are shown in Fig. 5. The best 1000 models from the neighbourhood algorithm occupy a much smaller range for the Sakarya Terrane and Armutlu Block examples. The broader range for the Istanbul Zone example shows that the data here provide weaker or possibly conflicting constraints on the model velocity profile. To improve the data misfit in such cases, a linearized inversion approach using *surf96* (Herrmann, 2013) is used to find an optimum model. Supplementary Fig. S7 shows the final fit of the dispersion curves calculated at each of the nodes shown in Fig. 5. The dispersion curves were calculated for the final S-wave velocity model, and compared to dispersion curves extracted from the Rayleigh and Love wave phase velocity tomography. Supplementary Fig. S7 also summarises the improvement in the misfit to the dispersion data provided by employing the linearised inversion (Herrmann, 2013) after the neighbourhood algorithm. Each node has a significant improvement in misfit following the linearized inversion ($> 50\%$).

## 3.4 Isotropic S-wave velocity maps

Fig. 6 shows depth slices through the final S-wave velocity model at depths of 1.5 km, 3.5 km and 5.5 km. The final S-wave velocity model is produced by performing a minimum curvature interpolation between our model nodes, which have the same spacing as our phase velocity model (Sec. 2.3). In the top 3 km of the crust we observe low S-wave velocities (1.6 - 2.0 km s$^{-1}$) on the north side of the northern fault strand, associated with the Adapazari basin and faulted sediments near Izmit. These low S-wave velocities are not observed at model depths of 3.5 km and below (Fig. 7), indicating that the Adapazari basin is likely not deeper than about 3.5 km. At 5.5 km depth, relatively low S-wave velocities (2.8 km s$^{-1}$) are clearly associated with the northern branch of the NAFZ, particularly within the zone of the Izmit rupture beneath Lake Sapanca at 40.7N and 30.2E. Faster S-wave velocities, up to 3.5 km s$^{-1}$, are observed within the Armutlu Block between the two strands of the NAFZ. As with the phase velocity maps, these high velocities are more prominent west of the Sakarya River to a depth of about 3.5 km. The slow velocities associated with the Pamukova basin along the southern branch of the NAFZ are much attenuated at 3.5 km depth, indicating that this basin is shallower than the Adapazari basin. We observe evidence in the southern part of the model for crystalline rocks below a depth of 1.5 km in the Sakarya Terrane, where S-wave velocities exceed 2.5 km s$^{-1}$. These high velocities are also observed in the far north of the model within the Istanbul Zone. Both the northern and southern branches of the NAFZ appear to exploit the regions where we observe high gradients in seismic S-wave velocity. Both branches of the main fault skirt the edges of the high velocity zone associated with the Armutlu Block.

## 3.5 Isotropic S-wave velocity vertical profiles

Fig. 7 shows two vertical sections through the S-wave velocity model along a North - South profile located at 30.2°E (profile A-A'), and 30.4°E (profile B-B'). In profile A-A' the low velocity zone associated with the heavily faulted sediments near Izmit (40.82°N) can be observed to a depth of $\sim$ 3.5 km, as can the Adapazari basin along the profile B-B'. In profile A-A' the Armutlu Block is clearly distinguishable as a region of high velocity ($\sim$ 2.8 km s$^{-1}$) extending towards the surface between 40.5°N and 40.6°N. It is clear that high velocity metamorphic rocks found in this region (Yılmaz et al., 1995) are located closer to the surface than the basement rocks of the Sakarya Terrane and Istanbul Zone. In both profiles, a zone of low velocity ($\sim$ 2.8 km s$^{-1}$) can be seen extending to a depth of at least 6 km beneath the location of the surface expression of the northern

branch of the NAFZ. This low velocity zone appears to be on the order of 10 km wide (40.65°N to 40.75°N). Low velocities associated with the southern branch of the fault zone are less clear, particularly for the eastern profile B-B', but are evident to 5 km depth beneath profile A-A'. However, it is difficult to distinguish the southern branch of the fault from the surrounding sedimentary cover of the Sakarya Terrane and Pamukova basin.

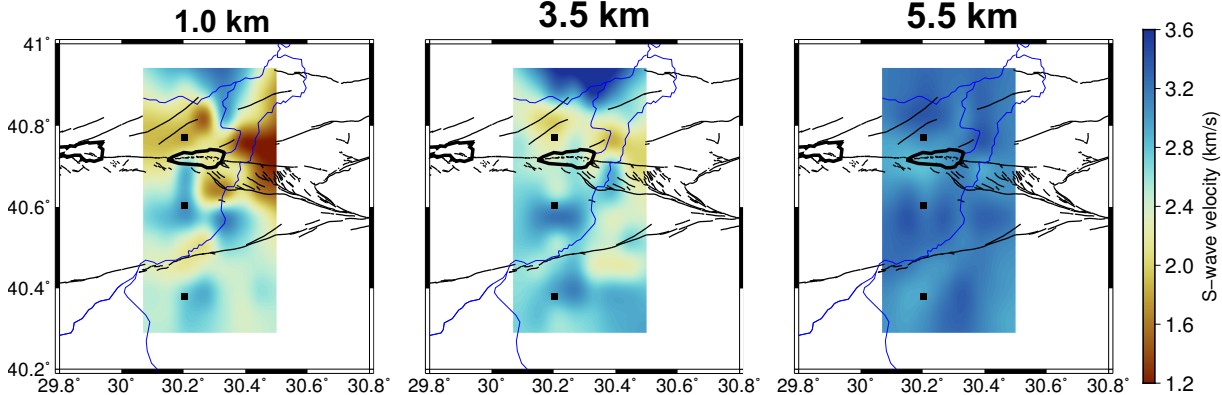

**Figure 6.** Isotropic S-wave velocity maps at 1.5, 3.5 and 5.5 km depth. Black lines show the mapped faults. The blue line represents the Sakarya River, flowing towards the north. The black squares represent the locations of the nodes shown in Fig. 5.

## 3.6   Azimuthal anisotropy

In order to quantify the level of azimuthal anisotropy in our phase velocity data set, we plot our raw phase velocity measurements against the azimuth of the propagation direction (from north). To reduce the scatter in the data and provide a meaningful measurement, we bin all of our phase velocity measurements by azimuth, with a bin size of 5°. The phase velocities within each bin are averaged to provide a mean measurement and a corresponding standard error. Rayleigh and Love wave observations are treated separately. Due to the presumed symmetry of propagation velocity in both directions between pairs of stations, our measurements are in an azimuth range of 0° to 180°. We attempt to fit the binned data at each period with the following function to describe the azimuthal variation of phase velocity (Smith and Dahlen, 1973):

$$c(\theta) = u_0 + A\cos(2(\theta - \phi_2)) + B\cos(4(\theta - \phi_4)). \tag{4}$$

where $u_0$ is the average (isotropic) phase velocity. $A$ is the amplitude of the $2\theta$ term, which describes an azimuthal variation with 180° periodicity. $\phi_2$ is the fast direction of the $2\theta$ term. $B$ is the amplitude of the $4\theta$ term, which has 90° periodicity, and $\phi_4$ is the corresponding fast direction.

The azimuthal variation of the raw Rayleigh wave phase velocity measurements between 2.0 and 8.0 s period is shown in Fig. 8. Fig. 9 shows the variation of fast direction and magnitude of anisotropy for all periods between 1.5 s and 10.0 s. Although there is considerable variability in the individual phase velocities, there is a robust dependence of phase velocity on propagation direction that is observed when averaging veocities in 5° azimuth bins. Fig. 9 shows a smooth variation in the fast

direction with increasing period of the wave. At short periods (2 - 3 s) the fast direction is aligned close to 90° from north, but changes smoothly to $\sim 50° - 70°$ from north above 5 s period. Below 2 s period, the anisotropy has a magnitude greater than 1%, but this magnitude decreases substantially between 2 and 4 s period, before increasing again at periods greater than 4.0 s to a value of $\sim 3\%$.

In general, the amplitude of the $4\theta$ term is at least 50% lower than the $2\theta$ term, which is to be expected for Rayleigh waves (Smith and Dahlen, 1973). The exception to these trends is at 2.0 s period. Here, the fast directions do not align with those observed at longer periods, and the $4\theta$ component has twice the amplitude of the $2\theta$ component. However, both the RMS misfit and the variance of the residuals between the observed data and eq. 4 are much greater at 2.0 s period, as is the case with the phase velocity tomography. In particular, the greater variance of the residuals implies a greater uncertainty in the data

fit. Greater variance in the 2 s phase velocities is likely due to the fact that waves at 2.0 s period are more sensitive to short wavelength heterogeneities near the surface.

     A further source of uncertainty in our calculation of azimuthal anisotropy is the unknown noise source distribution of the region. It is clear from the azimuthal distribution of our phase velocity measurements (Fig. S17 and Fig. S18 in the supplementary material) that there is a possible bias due to the number of ray paths that are oriented north – south. Fewer

observations are available for ray paths that are not aligned in the dominant direction, leading to higher uncertainty on our measurements of anisotropy. This effect is visible in Fig. 8: measurements taken from east – west oriented ray paths ($\sim 90°$) generally display a higher standard error of the mean than those for north – south oriented ray paths (0° or 180°).

     The azimuthal anisotropy of the Love wave phase velocities is shown in supplementary Fig. S13. The Love wave anisotropy is less clear. In general, the $2\theta$ fast direction lies between 25° and 40° from north. The $4\theta$ fast direction is more variable, mostly

lying between 85° and 120°. The average amplitude of the $2\theta$ term is 0.036 km s$^{-1}$. Whilst the amplitude of the $4\theta$ term is more comparable in amplitude to the $2\theta$ term than for the Rayleigh waves, it is still consistently smaller, with an average of 0.024 km s$^{-1}$. The RMS misfit and variance of the residuals is again higher at the shorter periods of 2.0 s and 4.0, again indicating sensitivity to shorter wavelength structural complexities near the surface. The azimuthal distribution of ray paths used in this analysis is shown in supplementary Figs. S14 and S15.

**4   Discussion**

**4.1   S-wave velocity model**

The horizontal resolution of the S-wave velocity model at depth in Fig. 7 is limited by the wavelength of the surface waves used in this study. Receiver function and autocorrelation studies of the region show that the shear zone associated with the NAFZ is perhaps no wider than $\sim 7$ km through the crust and into the upper mantle (Kahraman et al., 2015; Taylor et al., 2016). In the

upper crust, the main fault strands are estimated to be no more than a few kilometres wide in this region (Okay and Tüysüz, 1999). Low S-wave velocities associated with the northern branch of the NAFZ are observable in our model down to a depth of 6 km. Below this depth, we rely on observations derived from Rayleigh waves with a period greater than 8.0 s (phase velocity sensitivity kernels in supplementary Fig. S8). Assuming a phase velocity of 3 km s$^{-1}$, these waves have a wavelength of $\sim 24$

km. Thus, we cannot expect to resolve such a narrow structure at depth, unless it offsets rocks of differing seismic velocity. In the supplementary material (Fig. S9), we include the resolution kernels of the final S-wave velocity models at the three locations specified in Fig. 6.

Our tomographic models show that both the northern and southern branches of the NAFZ have exploited boundaries between major lithological units. In particular the metamorphic rocks of the Armutlu Block are clearly mapped due to the strong velocity contrast with rocks of the Istanbul Zone to the north and the Sakarya Terrane to the south (Figs. 3, 4 and 6).

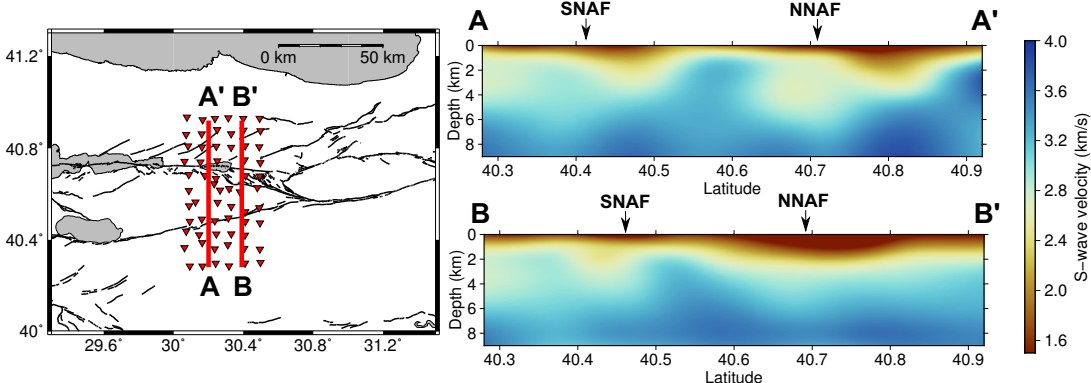

**Figure 7.** Top: Map of the Izmit-Adapazari region showing station locations of the DANA network as red triangles, and mapped faults as black lines. Thick red lines indicate the location of the vertical profiles taken through S-wave velocity model along lines A - A' and B - B'. Middle: Vertical S-wave velocity profile between A – A'. Bottom: Vertical S-wave velocity profile between B – B'. The profiles show S-wave velocity between the surface and 9 km depth. The approximate location of the surface traces of the northern and southern branches of the NAFZ are indicated by NNAF and SNAF, respectively.

Seismic velocity models of the crust in this region have also been constructed from teleseismic body wave tomography by Papaleo et al. (2017) and Papaleo et al. (2018). They image depth averaged seismic velocity between the surface and 90 km depth, with a vertical and horizontal resolution of $\sim$ 15 km (Papaleo et al., 2017, 2018). Despite the large difference in

model resolution and a non-overlapping depth range, Papaleo et al. (2017) and Papaleo et al. (2018) detect reduced crustal seismic velocities immediately to the north of the NAFZ, in the same regions we observe low S-wave velocities associated with the Adapazari Basin, and heavily faulted sedimentary cover in the north western part of the array (Figs. 6, 7). Low P-wave velocities observed by Papaleo et al. (2017) are also co-located with the low S-wave velocities detected in this study beneath the Pamukova basin. Papaleo et al. (2017) and Papaleo et al. (2018) also found relatively high seismic velocity at depth within

the Armutlu Block. We detect high S-wave velocities much closer to the surface, that we attribute to the shallow metamorphic rocks reported in this region (Yılmaz et al., 1995). We note that the relatively high seismic velocities we find in the upper crust of the Armutlu Block also corresponds to a region of relatively low electrical resistivity found by Tank et al. (2005) in the upper 10 km.

The depth of sedimentary cover of the Adapazari basin has been estimated to be at least 1.0 km in some locations (Komazawa

et al., 2002). These estimates were made by inverting Rayleigh wave phase velocity measurements from microseisms recorded

at two arrays within the basin. Due to a lack of measurements below 0.6 Hz ($> \sim 1.6$ s period) the inversion of Komazawa et al. (2002) assumed an S-wave velocity of 3.5 km s$^{-1}$ below a depth of 500 m in the basin. Our velocity model, which incorporates Rayleigh wave observations up to 10.0 s period, indicates that S-wave velocity may be no greater than 3.0 km s$^{-1}$ up to a depth of 2.5 km within the basin. Our measurements therefore imply that the Adapazari basin could have a depth of up to 2.5 km based on the observed increase in S-wave velocity at this depth. Similarly, the Pamukova basin may be as deep as 2.5 km, though it is difficult to accurately detect the depth to material interfaces using only surface wave observations.

Studies of the near surface structure of the San Jacinto fault zone in southern California (Allam and Ben-Zion (2012); Zigone et al. (2015)) observe prominent 'flower structures' associated with the fault. These structures are zones of low seismic velocity that are wide near the surface, become narrower with depth, and are interpreted to be a damage zone created during fault propagation through undeformed crust. The surface wave analysis does not enable us to observe a narrowing with depth of the low-velocity zone associated with the northern branch of the NAFZ in Fig. 7. Nonetheless the low velocity anomalies associated with the Adapazari and Izmit regions might be interpreted as crust that has been damaged by movement on and around the northern strand of the fault. It is clear that the strongest contrasts in seismic velocities in our model (Figs. 3, 4 and 6) are associated with boundaries between the three main tectonic units. The North Anatolian Fault Zone appears to have developed along pre-existing tectonic boundaries.

Such seismic velocity contrasts across an active strike-slip fault are also present in California on the creeping section of the San Andreas Fault to the north of Parkfield where the fault trace is located along a strong seismic velocity contrast between the Great Valley sedimentary sequence and the granites of the Salinian terrane (Eberhart-Phillips and Michael, 1993; Thurber et al., 2006). This phenomenon is also observed across the Hayward fault near San Francisco where there is a clear seismic velocity contrast between the Great Valley sequence and the Franciscan Complex (Hardebeck et al., 2007; Thurber et al., 2006). Eberhart-Phillips and Michael (1993) suggest that the San Andreas Fault is likely to creep in sections where this clear velocity contrast exists, whilst being locked and rupturing seismogenically where the velocity contrast across the fault is less defined. However, this association between a creeping fault segment and a clearly defined velocity contrast evidently does not hold for this section of the NAFZ where the 1999 Izmit and Düzce earthquakes occurred. Furthermore, a recent geodetic study found evidence of only low creep rates on this segment, probably related to earthquake after-slip at shallow depths (Hussain et al., 2016).

The relatively high S-wave velocities we observe within the Armutlu Block likely indicate metamorphic rocks and pre-Jurassic basement (Akbayram et al., 2016) of which the surface outcrops are of unknown provenance and age (Okay and Tüysüz, 1999). This metamorphic unit within the Armutlu Block is evidently resistant to strain, which is deflected onto the northern and southern branches of the NAFZ that bound this high S-wave velocity region. This behaviour is also observed in the near surface structure of the south eastern section of the Alpine Fault on South Island, New Zealand, where the fault trace is located at the edge of the metamorphic Haast Schist, and cuts through thick coastal sediments (Eberhart-Phillips and Bannister, 2002). Fichtner et al. (2013) image the S-wave velocity structure of the upper mantle beneath the NAFZ using full waveform inversion. At this much larger length and depth scale, they also note that the NAFZ appears to be bounded by tectonic blocks

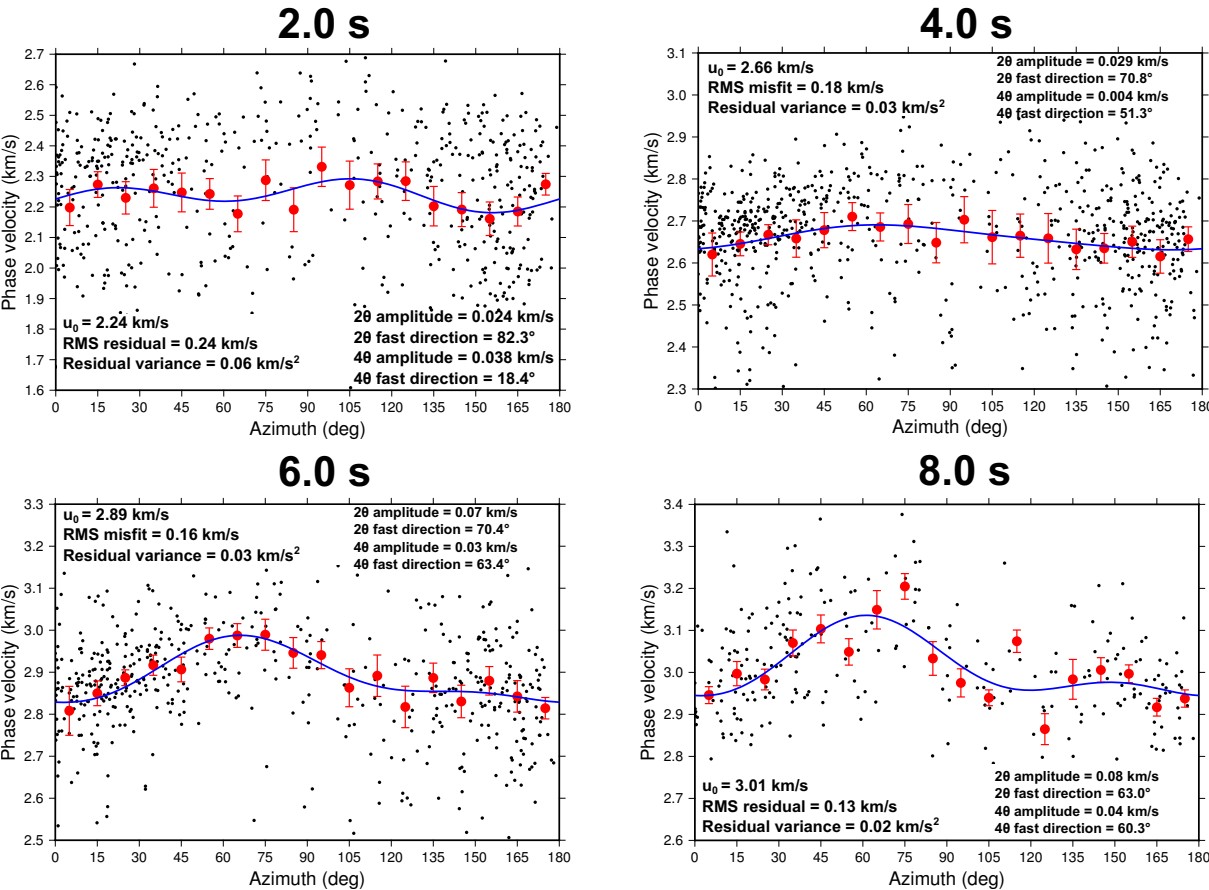

**Figure 8.** Azimuthal variation of Rayleigh wave phase velocities with propagation azimuth (from north). Black dots indicate the raw phase velocity measurements, large red dots show the average of the phase velocities within 5 degree azimuth bins, and the corresponding standard error of the mean for the bin. The blue line is the best fitting curve (eq. 4) to the binned data (red dots). $u_0$ is the average (isotropic) phase velocity. We show the root mean square misfit of the blue curve to the phase velocity measurements, as well as the variance of the residuals. We indicate the $2\theta$ and $4\theta$ amplitudes and fast directions that correspond to the blue curve. The azimuthal distribution of ray paths used in this analysis is shown in supplementary Fig. S14.

of high seismic velocity. They interpret this as evidence that the fault zone developed along the edges of high-rigidity blocks, analogous to our observations for the near-surface structure of the Armutlu Block.

### 4.2 Azimuthal anisotropy

The $2\theta$ and $4\theta$ fast directions for Rayleigh waves varies between 50° - 90° from north (Fig. 9), whilst Love wave $2\theta$ fast directions vary from 20° to 40° from north (Fig. S13). The Love wave $4\theta$ fast direction is highly variable, with no distinct pattern that can be readily observed.

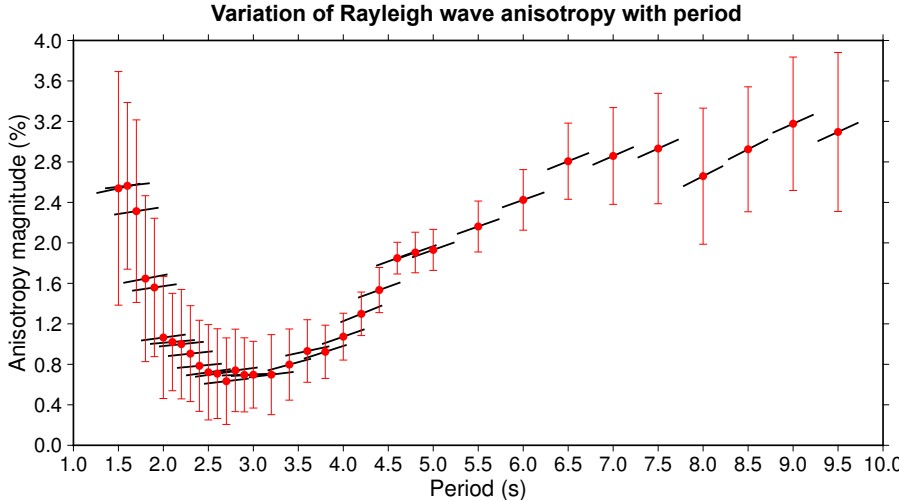

**Figure 9.** Variation of $2\theta$ Rayleigh wave anisotropy with period in the Izmit-Adapazari region. The red dots are the measured magnitude of anisotropy at each period, and the corresponding uncertainty is the standard deviation of the anisotropy magnitude taken from the covariance matrix during the curve fitting process described in section 3.6. The black lines indicate the angle from north of the $2\theta$ fast direction at each period, where the top of the plot represents north.

Our observations of azimuthal anisotropy are complementary to the observations of previous studies along the North Anatolian Fault. Two studies of shear wave splitting measurements of the Karadere - Düzce segment ($\sim$ 50 km east of the current study region) by Peng and Ben-Zion (2004) and Peng and Ben-Zion (2005) also display a seismic fast direction in the upper crust that clusters between 45° and 90° from north, often aligning parallel to the strike of the North Anatolian Fault. Further

shear wave splitting measurements made by Hurd and Bohnhoff (2012) at the station CAY, located within our study region to the east of Lake Sapanca (Fig. 1), also showed directions between 30° and 90°, with the majority falling between 40° and 50°. Further east, the fast polarisation directions measured by Hurd and Bohnhoff (2012) are more commonly aligned NW – SE.

There are two possible mechanisms of crustal anisotropy: stress-controlled or structure-controlled. If the anisotropy is stress-controlled, it is expected that the the fast direction will align with the direction of maximum horizontal compression in the

10 stress field due to the closure of cracks on the perpendicular direction (Crampin and Lovell, 1991). For an east – west striking fault, this would result in an expected fast direction aligned NW – SE, or 120° – 160° from north (Bohnhoff et al., 2006). Our observations, and those of previous studies (Peng and Ben-Zion, 2004, 2005), show that this is not the case, at least for stations located close to the fault. A dominant fast direction between 50° – 90° (NE – SW) from north (Fig. 9) indicates that the anisotropy in the region is likely structure-controlled. This observation was also noted in anisotropic receiver functions

by Licciardi et al. (2018), who found that the fast shear wave polarisation directions along the central portion of the North Anatolian Fault align with the strike of mapped faults at stations located close to those faults, implying structure-controlled anisotropy.

Fig. 9 shows a nearly 90° fast direction at 2 – 3 s period ($\sim 0 - 3$ km depth) that aligns approximately with the strike of the North Anatolian Fault through the region. This observation clearly implies structure-controlled anisotropy that is dominated by faulting in the very upper crust, similar to the observations of Licciardi et al. (2018) for the top 15 km of the central section of the North Anatolian Fault. At periods greater than 4.5 s (Fig. 9), our observed fast direction does not systematically align with any of the mapped faults in the region (Fig. 1). Instead, the fast direction at these periods is better compared to the 45° direction of maximum extension for the Izmit-Adapazari region calculated from interseismic GPS data by Allmendinger et al. (2007), and is consistent with shear wave splitting measurements from the central portion of the North Anatolian Fault made by Biryol et al. (2010), who found a fast polarisation direction that varied between 35° and 60°. Further analysis of shear wave splitting results by Vinnik et al. (2016) show an average fast direction of $\sim 60°$ down to a depth of about 30 km.

This close correspondence between the seismic fast direction and the direction of maximum extension implies that the structure-controlled anisotropy is the result of mineral foliation within the crust. Some minerals in upper crustal rocks, such as micas and amphibole, typically have cleavage planes or crystallographic axes aligned with the dominant strain direction, and are the dominant source of anisotropy within the bulk rock (e.g. Kern and Wenk (1990), Mainprice and Nicolas (1989), Sherrington et al. (2004)). These minerals are particularly common in high grade metamorphic rocks such as slates and schists, and are likely abundant within the Armutlu Block. Analysis of samples of calcite and amphiboles taken from the Uludag Massif ($\sim 100$ km south-west of Izmit-Adapazari) by Farrell (2017) show that the fast propagation for both P and S waves aligns parallel to the foliation direction in these minerals. We therefore think it likely that the seismic fast directions we observe at longer periods are determined by deformation fabrics aligned with the dominant shear regime.

## 5 Conclusions

We utilised the ambient noise field recorded at a temporary network in the Izmit-Adapazari region of north western Turkey to retrieve Rayleigh and Love waves propagating between the stations of the array. We performed surface wave phase velocity tomography, followed by an inversion for S-wave velocity structure, with waves of periods from 1.5 to 10.0 s to image the shear wave velocity in the top 10 km of the North Anatolian Fault Zone.

Our model shows low S-wave velocity to the north of the NAFZ, associated with faulted marine clastic sediments near Izmit (Akbayram et al., 2016) and with the Adapazari sedimentary basin, which we estimate to have a thickness of at least 2.5 km. In between the two branches of the NAFZ, we observe a high velocity region linked to metamorphic and igneous rocks in the Armutlu Block. It is likely that this high S-wave velocity in the upper crust is indicative of a rheologically strong region that preferentially localises strain at the boundaries of the Armutlu Block, particularly along its northern boundary which has been identified as the Intra-Pontide Suture Zone. We also image the Pamukova basin as a region of low S-wave velocity to a depth of about 2.5 km, associated with the southern branch of the NAFZ. Both basins are likely related to pull-apart motion along the northern and southern branches of the NAFZ, where they are oblique to the principal shear direction.

To the south of the NAFZ, we image the Sakarya Terrane as a region of moderate to high S-wave velocity, consistent with the Sakarya Terrane being an accretionary complex of sedimentary rocks overlying a metamorphic crystalline basement

(Yılmaz et al., 1995). Our analysis of the azimuthal variation in phase velocities finds that regional seismic anisotropy is likely structure-controlled. At short periods, both Rayleigh and Love waves have a fast direction which roughly aligns with the strike of the North Anatolian Fault (east – west), as opposed to the direction of maximum compression (NW – SE). At longer periods (> 4.0 s), the fast direction smoothly transitions from the maximum shear direction towards the principle extension direction of the lithosphere (NE – SW), indicating that mineral fabric may be the source of azimuthal anisotropy. Studying the relationship between the three distinct tectonic units of the region, including the patterns of seismic anisotropy, provides insight into the potential for strain localisation along both the northern and southern branches of the NAFZ. This knowledge is critical to understanding the long term behaviour of the fault zone, and the seismic hazard that it poses.

*Data availability.*  The final S-wave velocity model of the Izmit-Adapazari region is included as an ASCII text file within the supplementary material. Data for this study can be found at the IRIS Data Management Centre under network code YH (2012 - 2013) (DANA, 2012).

*Competing interests.*  The authors declare that they have no conflict of interest.

*Acknowledgements.*  G. Taylor is supported by the Leeds-York Doctoral Training Partnership of the Natural Environment Research Council (NERC), UK. G. Hillers acknowledges support through a Heisenberg fellowship from the German Research Foundation (HI 1714/1-2). The DANA array was part of the Faultlab project, a collaborative effort by the University of Leeds, Kandilli Observatory and Earthquake Research Institute, and Sakarya University. Major funding was provided by the UK NERC under grant NE/I028017/1. Equipment was provided and supported by the NERC Geophysical Equipment Facility (SEIS-UK) Loan 947. We would like to thank S. Schippkus and an anonymous reviewer for their detailed reviews that helped improve the manuscript.

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
