# Peer review of "Near surface structure of the North Anatolian Fault Zone from Rayleigh and Love wave tomography using ambient seismic noise."

_Solid Earth, 2018_

## Referee Comment (RC1) · S. Schippkus (Referee) · 6 Nov 2018

**General Comments**

In this study, the authors present a shear velocity model of the North-Anatolian Fault Zone near 30.3°E longitude for the top 10km with an extent of 30x70km. The model was computed using a classical surface wave tomography approach with surface waves extracted from ambient noise cross correlations. It provides a detailed image of the studied region that matches observations from previous studies and seems to bring new insight into the local geological structure, including previously unavailable esti-mates of sedimentary basin depths. I am not familiar with the study region and therefore will focus this review on the technical/methodological aspects.

The work has been done in a professional manner and is mostly well documented, although some aspects remain unclear. Many details about the method are given, but some essential ones are missing that would help the reader better understand the approach, reasoning behind it, and assess the confidence in the interpretation of the model.

I think this work presents interesting observations for the region and ambient noise in general, but needs more work on the methodological part to make it fully reproducible and comprehensible. Therefore, I suggest this work to be published after moderate to major revision.

**Specific Comments**

**Major Points**

1. The interaction of group-velocity measurement and phase-velocity measurement is a bit unclear. The group velocity measurements are described well, but how exactly they are used to constrain the phase velocities and how the phase velocities are measured is not.

2. The benefit of the 2-step approach to shear-velocity inversion (neighbourhood algorithm to find initial model for linearised inversion) is not explicitly demonstrated or referenced. The linearised inversion does decrease the misfit significantly using the found initial model, but how does it compare to a 'guessed' initial model? I assume the authors did some tests, encountered problems, or have previous experience with linearised inversions which lead to deciding on this procedure. I would appreciate more insight into the reasoning, because this approach seems to have potential to help with the choice of an initial model for linearised inversions in general.

3. The checker-board test provides only rudimentary insight into the resolution of the phase velocity maps, because the results of only one single velocity-distribution, which

is not argued for, are presented. See Lévêque et al., 1993 on how checkerboard tests can lead to misinterpretations, if not done carefully. Because the inversion algorithm used to construct the phase-velocity maps does not provide an inherent resolution estimation (to my understanding), I suggest to add additional checkerboard tests to better judge the ability to image features of different sizes and magnitudes. On a related note, can the authors share insight on how lateral resolution estimates may translate from phase-velocity maps to shear-velocity maps?

> Lévêque, J.-J., Rivera, L. & Wittlinger, G. On the use of the checker-board test to assess the resolution of tomographic inversions. Geophys. J. Int. 313–318 (1993).

4. What is the depth resolution of the shear velocity inversion? The linearised inversion scheme implemented in CPS (Hermann 2013) provides a resolution matrix as output. I suggest to add a figure of example resolution matrices to the Supplementary (maybe 3 matrices for the 3 previously shown nodes, or a mean resolution matrix of all nodes) to give the reader a better understanding of the validity of the author's interpretation of the model. This would be in addition to the "vertical resolution" insight gained from the depth sensitivity kernels as the resolution matrix better illustrates the interdependence of different depths, possibly giving insight into potential biases in the final model and interpretation thereof.

5. The authors investigate and interpret the azimuthal anisotropy found by comparing Love- and Rayleigh-wave maps. I am curious to see if the authors also investigated the potential bias in the group- and phase-velocity measurements themselves that may be introduced e.g., by an inhomogeneous noise source distribution in the study region. Are there other studies for the region investigating the noise source distribution and the effect this may have on velocity measurements?

6. Why did the authors chose to not invert the group velocities for group-velocity maps? The bulk of the work is already done by manually measuring all dispersion curves. The methodology would be very similar to the approach based on phase velocities and

using the group velocities may provide additional insight or help better constrain the imaged features.

7. The used colormap excels at pronouncing differences in the models (Figures 3, 4, 5, 6, 7, S9, and S10), but can lead to misinterpretation, because it is not perceptually uniform and introduces visual discontinuities where there are no discontinuities in the data. I suggest to use another colormap that is perceptually uniform. One collection can be found here: http://www.fabiocrameri.ch/colourmaps.php. Why non-uniform colormaps can be problematic, see e.g.,

> Michelle A. Borkin and Krzysztof Z. Gajos and Amanda Peters and Dimitrios Mitsouras and Simone Melchionna and Rybicki, Frank J. and Charles L. Feldman and Hanspeter Pfister. (2011) Evaluation of Artery Visualizations for Heart Disease Diagnosis, IEEE Transactions on Visualization and Computer Graphics 17, 12.

8. The axis labels on Figures 1a, 1b, 3, 4, 6, and 9 are incomplete/wrong. Especially figure 9 is hard to read and understand like this. Similarly, the colorbar labels on Figures 3 and 4 are missing punctuation marks. The colobar label on Figure 6 is unreadable.

**Minor Points**

**Main Text**

- Page 2, Line 19-20 / Figure 1: How high is high topography? A colorbar for topography would be helpful in the figure. This would also help to judge whether there may be possible bias in the measurements caused by station altitudes.

- Page 2, Line 23: Please give a reference for the observations of "Striations and down dip motion on faults".

- Page 4, Lines 26-33 and Figure 2: The authors mention several possible explanations for the near-0 arrivals that are dominant on the ZZ-component, which indeed is commonly observed. Do these also explain the multiplets of horizontal lines around -10s and +10s in the ZZ-panel or do these require another interpretation? Similar features

have been observed in Lehujeur et al., 2018 that have been found to be instrument artefacts caused by the used digitisers.

> Lehujeur M., J. Vergne, J. Schmittbuhl, D. Zigone, A. Le Chenadec & EstOF Team (2018). Reservoir imaging using ambient noise correlation from a dense seismic network. J. Geophys. Res., https://doi.org/10.1029/2018JB015440.

- Page 6, Lines 5-8: The 13.5km inter-station distance threshold makes sense for measurements at 1.5s period. In this study, periods up to 10s are used, which would then lead to all station pairs with less 30km (assuming c=3km/s) inter-station distance at 10s. Instead, station-pairs are excluded based on visual inspection for all periods > 1.5s. Why did the authors choose a different approach for 1.5s and all other periods?

- Page 6, Lines 14-16: This part needs more details to be fully reproducible. I understand that the theoretically computed phase-velocity dispersion curves (from group-velocity curves) are used to constrain the phase-velocity measurements. How are the computed phase-velocity curves constructed? How are the phase velocities measured, i.e., what part of the waveform is used (e.g., zero-crossings, instantaneous phase, . . .)? How exactly does the theoretical phase-velocity curve constrain the measured phase velocities? Are the phase velocities picked manually or automatically?

- Page 6, Line 20-22: Mainly because it is not well described how exactly phase velocities are picked, this part is a bit confusing. Depending on the measurement procedure, it may be going in circles as follows: 1) Measure phase travel-time (if measuring e.g., zero-crossings) 2) Compute phase-velocity estimated from inter-station distance and great circle propagation 3) Convert to travel times. Please clarify this section.

- Page 7, Line 29: "A total of 20050 . . . for each node. . . ". Does this mean 20050 times the number of nodes (however many that may be) or 20050 distributed over all nodes?

- Page 7, Line 30: Eq. (2) is referenced before being introduced (in the very next sentence). Maybe change the order of these two sentences.

[Figure]

- Page 9, Lines 7-9: Did the authors try using the regional Karahan et al., (2001) model (see Page 6, Lines 16-17) as the initial model for the linearised inversion? Please elaborate on the benefit of using the initial model obtained from the neighbourhood algorithm. Why does the neighbourhood algorithm not converge to the same solution as the linearised inversion and how different are the final models retrieved from neighbourhood and linearised inversion?

- Page 14, Lines 15-19 / Figure 9: This section is a bit hard to follow, mainly because the axis labels in Figure 9 are unreadable.

**Supplementary**

- Text S4, Figure S4: This looks similar to a standard L-curve analysis, where a regularization parameter is usually chosen near the maximum curvature of the model-variance-vs.-parameter curve (e.g., Hansen & O'Leary, 1993). This is not mentioned explicitly, though. Is this choice of dampening based on maximum curvature or on subjective judgement alone?

> Hansen, P. C. & O'Leary, D. P., 1993. The Use of the L-Curve in the Regularization of Discrete Ill-Posed Problems, SIAM Journal on Scientific Computing, 14(6), 1487–1503.

- Text S10 and Figures S14, S15: There seems to be a slight bias of available interstation-azimuths that can not be explained by the N/S-dominant station distribution alone. The Rayleigh wave distributions (Fig. S14) are dominated by slightly NNE/SSW-rays (especially at 6s and 8s), while the Love wave distributions (Fig. S15) are dominated by slightly NNW/SSE-rays (well visible at all periods). Do the authors have an explanation for this difference? Maybe this is a sign for bias introduced during the visual inspection and selection of dispersion curves (main text: page 6, line 9) that could be caused by a potential difference in noise source distribution for Love and Rayleigh waves (See e.g., Riahi et al., 2013).

On that note, I suggest to add to Text S10 why there are generally less paths available

for 2.0s than 4.0s, as this is not explained by the (in the main text) wavelength-based exclusion alone and a consequence of the visual inspection, I assume. Do you find dominance of higher modes at shorter periods that lead to these periods being preferentially excluded?

> Riahi, N., G. Bokelmann, P. Sala, and E. H. Saenger (2013), Time-lapse analysis of ambient surface wave anisotropy: A three-component array study above an underground gas storage, J. Geophys. Res. Solid Earth, 118, 5339–5351, doi:10.1002/jgrb.50375.

**Technical Corrections**

**Main Text**

- Figure 1: Axis labels for both maps are unreadable. I suggest to add a colorbar for topography. I did not find the station names being used in the text, therefore the authors could remove the explanation.

- Figure 3: Axis labels for all maps are unreadable. Colorbar labels have no punctuation marks and colorbar has no title (phase velocity).

- Figure 4: Same as Figure 3.

- Figure 6: Same as Figure 3 & 4. Additionally, the colorbar labels are incomplete.

- Figure 9: Same as Figure 3, 4, 6

**Supplementary**

- Figures S1, S2: Both figures are a bit low quality.

- Figure S13: "The blue line is the best fitting curve the raw data." -> "The blue line is the curve best fitting the raw data."

---

## Referee Comment (RC2) · Anonymous Referee #2 · 8 Nov 2018

I think that the need for the present work is indisputable given that the high seismic hazardous potential of the study area has, but I got an impression that this work lacks some very interesting points which can justify the quality of the work when reading it. In that respect authors seem to take some critical issues explained below quite superficially. I think this work requires a major revision prior to possible publication in the journal of Solid Earth. At this stage I strongly suggest authors seriously consider following issues listed below.

General Comments

Although they emphasize the importance of good and reliable knowledge on crustal

structures along the continental shear deformation zones at the very beginning in the introduction, and since this is one of the primary task for taking all such efforts in the region, I am very upset why they avoid to interpret their results mainly around this target which could be vitally important for future studies that aim at a decent seismic scenario for the region.

There have been numbers of recent geophysical model and observations in a region including the study area and further west dealing with the branch of the NAFZ beneath the Sea of Marmara. However, introduction significantly lacks of a compilation of previous studies and their findings including the DANA experiment.

Isotropic velocity In the present work, inversion results for shear wave velocity for deeper sections at 3.5 and 5.5 km do not provide profound velocity contrasts among three tectonic zones, namely, Istanbul, Armutlu-Almacik, and Sakarya Zones (see Fig. 6) whereas using the same network and teleseismic P-and S arrivals Papaleo et al. (2017, 2018) were showing clear separation reflected as relatively high wave speeds beneath Istanbul Zone to the north, and low beneath Sakarya Zone to the south that is mostly likely due to the lithological differences down to, at least, the depth of 20 km.

Only for the first 1.5 depth range, resolution is sufficient to resolve shear zones along the northern branch. There down to the depth of 1.5 km major difference is claimed by the authors to be associated with low S-wave velocity to the north of the NAFZ, associated with faulted marine clastic sediments near Izmit (Akbayram et al., 2016) and with the Adapazari sedimentary basin. I think a detailed introduction with more geological constraint as well as other geophysical data to support this and further velocity variations at this depth range is missing. Such introduction is crucial since below this range velocity variation does not show high resolution details.

It seems there is an effect of N-S elongated azimuth of station pairs on resolved images. This effect can be investigated using sensitivity analysis, i.e., checkerboard test results. I am aware that authors have already added materials in Supplementary but

[Figure]

I believe it is much better if given within the Sensitivity Analysis section of the main text. In this way, later they can use this by putting quantitative arguments when they describe the results (reliability of various features which will be potentially examined in the Discussion).

I would like to see the ray-paths of periods and their checkerboard results in supplementary file to be able to see the influence of dominance of N-S orientation of station-pairs in your data set.

Extraction of surface wave velocities

→ According to my recollection, in some studies dealing with ambient noise inversions in the literature, group velocities and related time information are used for further inversion process. Here authors are using phase velocities. Perhaps this has to be addressed in the text.

→ Figure 2 is interesting. One of the first things that is prominent on this figure is the zero-offset energy. What might be the major source for that? Needs to be clarified.

Azimuthal anisotropy Large scatter azimuthal variations of phase velocities (see Fig. 8 & S13) under the presence of N-S dominating azimuth of station-pairs. Thus long period behavior of directional dependent phase velocities is doubtful. And thus, a frequency varying fast velocity directions (with increasing uncertainties as period increases) is also not too convincing.

This work examines anisotropy issue with a superficial discussion regarding early constraints on seismic anisotropy in the region. Authors appear to take the discussion regarding seismic anisotropy only using a single SKS splitting study (Biryol et al., 2010), which has been informative for upper mantle anisotropy. However, there are a few earlier studies performed along the NAFZ (central and western NAFZ) with direct observation of crustal anisotropy. No specific discussion in the light of earlier works revealing upper crustal anisotropic structure mainly based on shear wave splitting structure (e.g.

Peng and Ben-Zion, 2004-2005; Hurd and Bohnhoff, 2012) or entire crust from RFs analyses (Vinnink et al., 2015; Licciardi et al. 2018).

The question on what part(s) of the area may indicate structure-induced, and what part(s) stress-induced anisotropy is still ambiguous. Moreover, a single model for such a complicated tectonic setting with significant lateral heterogeneities cannot be represented a single-smooth depth-varying model with very consistent SKS orientations (see e.g. Peng and Ben-Zion, 2004-2005; Hurd and Bohnhoff, 2012; Vinnink et al., 2016). At least early shear wave splitting and RFs data suggests the opposite what the current work says.

Another thing I could not figure out is that authors do not provide any clue regarding radial anisotropy? If they are already able to invert both love and Rayleigh wave wouldn't it be possible to visualize radial and tangential shear wave speed variations at various depth?

More importantly, I am seriously wonder why they have not gone for a detailed harmonic analysis that can provide depth variation of fast polarization azimuths on a finer spatial resolution using on available data set.

I would omit this part unless it is supported with a more convincing and detailed analysis of the data set.

Figures For Figs. 1, 3, 4, and 6, values of latitude and longitude is strange.

References

Two references of Şengör (Şengör and YÄślmaz, 1981; Şengör et al., 2005) are not listed in the alphabetical order.

---

## Author Comment (AC1) · 6 Dec 2018

Dear Sven Schippkus,

We would like to thank you for taking the time to produce a detailed review of our manuscript. We have taken each of your comments into consideration, and you can find our responses below. Any page and line numbers refer to the clean, updated manuscript. I will shortly upload the 'tracked changes' version of the manuscript, once it has been checked and finalised.

**Major points**

[Figure]

The interaction of group-velocity measurement and phase-velocity measurement is a bit unclear. The group velocity measurements are described well, but how exactly they are used to constrain the phase velocities and how the phase velocities are measured is not.

We realise that the current description of how the phase velocities are determined is unclear. A description can be found in the documentation on the *do_mft* program that we use to determine phase velocity (Herrmann et al., 2013). In particular, phase velocities are calculated using the analytic signal of a narrowly bandpassed surface wave using the equation:

$$c = \frac{\omega_0 r}{-\Phi + \frac{\pi}{4} + \frac{\omega_0 r}{U_0} + N2\pi} \tag{1}$$

where $\phi$ is the instantaneous phase of the surface wave, $\omega_0$ is the centre frequency of the filter, $r$ is the inter-station distance and $U_0$ is the group velocity. $N$ is some integer. Thus, once the group velocity curve is known, the corresponding phase velocity curve(s) can be calculated. The ambiguity in the phase velocity curves arises from the factor $N2\pi$ in this equation. This where the a priori earth model is used. *do_mft* uses eq. 1 to calculate a suite of phase velocity curves for various values of $N$, and the curve that most closely matches the synthetic phase velocity curve of the a priori Earth model is considered to be the one that corresponds to the correct value of $N$. This phase velocity curve is picked manually in our case.

2. The benefit of the 2-step approach to shear-velocity inversion (neighbourhood algorithm to find initial model for linearised inversion) is not explicitly demonstrated or referenced. The linearised inversion does decrease the misfit significantly using the found initial model, but how does it compare to a 'guessed' initial model? I assume the authors did some tests, encountered problems, or have previous experience with linearised inversions which lead to deciding on this procedure. I would appreciate more

insight into the reasoning, because this approach seems to have potential to help with the choice of an initial model for linearised inversions in general.

The main benefit of using the neighbourhood algorithm in this case is to provide a much broader overview of the acceptable parameter space of the inverted S-wave velocity models, rather than just presenting one 'best-fitting' model from a linearised inversion. In addition, presenting the results of the neighbourhood algorithm such as in the current Fig. 5 also allows the reader to form an intuitive (if only qualitative) understanding of potential uncertainties in the S-wave velocity model. However, limitations in the number of free parameters that can be efficiently inverted for using the neighbourhood algorithm (∼30) can cause problems. This is best demonstrated by the most northern model node in Fig. 5, which displays a larger range of acceptable models and a considerably higher misfit than the other two nodes presented. This may in part be due to the rather coarse parameterisation imposed on the problem by the neighbourhood algorithm, which is unable to provide a satisfactory fit to the dispersion data collected within the sedimentary basins that are located in that part of the model.

The 2-step inversion process has been applied in fault zone imaging before (Hillers and Campillo, 2018), and is an attempt to compromise between these two problems. We want to provide the reader with an overview of the acceptable parameter space and an intuitive sense of the uncertainty in different parts of our model through the neighbourhood algorithm, as well as presenting a model that provides the best fit to the data. As the reviewer has pointed out, the improvement of fit following the linearised inversion is substantial, and as the neighbourhood algorithm results are already to hand, they may as well be used as the initial model to guide the linearised inversion, rather than using a guessed model, or the Karahan model (which is very coarsely parametrised).

We have updated the manuscript to include the full justification of our approach as outlined above. This information can now be found on page 8, lines 27 – page 9 lines 1 – 2.

> Hillers and Campillo, 2018. Fault zone imaging from correlations of aftershock wave-forms. *Pure. Appl. Geophys.* 175.

3. The checker-board test provides only rudimentary insight into the resolution of the phase velocity maps, because the results of only one single velocity-distribution, which is not argued for, are presented. See Leveque et al., 1993 on how checkerboard tests can lead to misinterpretations, if not done carefully. Because the inversion algorithm used to construct the phase-velocity maps does not provide an inherent resolution estimation (to my understanding), I suggest to add additional checkerboard tests to better judge the ability to image features of different sizes and magnitudes. On a related note, can the authors share insight on how lateral resolution estimates may translate from phase-velocity maps to shear-velocity maps?

We agree with the reviewer's concerns regarding the issues with checkerboard tests. However, as the reviewer stated, the inversion algorithm we used for the phase velocity tomography does not include an inherent resolution estimate. In such cases, checkerboard tests remain the standard tool for estimating horizontal resolution in tomography studies.

We do not believe that including further checkerboard tests would provide the reader with any more useful information on lateral resolution in this case. Displaying a fixed checkerboard wavelength at different wave frequencies, as we currently do, covers the same physical parameter space as showing different checkerboard wavelengths at a fixed frequency, and makes adding further checkerboard wavelengths redundant.

Nonetheless, the reviewers point is valid, and we do wish to provide evidence of the robustness of our tomography to recovering anomalies of different amplitudes and shapes. We have extended the supplementary information to include two new resolution tests for our ray path distribution at 4 s period: we show the recovery of two spikes (of different amplitude to the checkerboard) in Fig. S12, and in Fig. S13 we show the ability of our tomography to recover a randomly generated velocity model

that contains anomalies of differing wavelengths. We hope that these further tests provide greater confidence in the horizontal resolution of our phase velocity tomography. Furthermore, we now provide the details of the lateral interpolation performed when constructing the final S-wave velocity model on page 12, line 7 - 9.

4. What is the depth resolution of the shear velocity inversion? The linearised inversion scheme implemented in CPS (Hermann 2013) provides a resolution matrix as output. I suggest to add a figure of example resolution matrices to the Supplementary (maybe 3 matrices for the 3 previously shown nodes, or a mean resolution matrix of all nodes) to give the reader a better understanding of the validity of the author's interpretation of the model. This would be in addition to the "vertical resolution" insight gained from the depth sensitivity kernels as the resolution matrix better illustrates the interdependence of different depths, possibly giving insight into potential biases in the final model and interpretation thereof.

In the supplementary material (Fig. S9), we have added the requested resolution kernels for the 3 model nodes displayed in Fig. 5 of the main text.

5. The authors investigate and interpret the azimuthal anisotropy found by comparing Love- and Rayleigh-wave maps. I am curious to see if the authors also investigated the potential bias in the group- and phase-velocity measurements themselves that may be introduced e.g., by an inhomogeneous noise source distribution in the study region. Are there other studies for the region investigating the noise source distribution and the effect this may have on velocity measurements?

The analysis performed in Fig. 8 is based upon the raw phase velocity measurements, with the Rayleigh wave (main text) and Love wave (supplementary) phase velocity measurements treated separately. The raw phase velocity measurements are represented as black dots in the figures. The raw data is binned and averaged in 10 degree azimuth bins (red dots) in order to clean up the scatter in the measurements so that a first order analysis of the azimuthal variation of the phase velocities can be performed.

Whilst we do not know of any study that specifically investigates the noise source distribution in this region, it is likely that the noise is predominantly aligned perpendicular to the coastlines of the Black Sea and the Mediterranean. Evidence of this is visible in Figs. S17 and S18, giving a first order overview of the azimuthal distribution of our phase velocity measurements, which are dominated by north-south oriented paths. The main effect of the presence of an anisotropic noise source distribution is to increase the uncertainty of our measurements taken from azimuths that are not aligned in the dominant direction (e.g. east-west), as fewer total measurements are available. This effect is visible in Fig. 8 for example: the binned data at azimuths $\sim$ 90 degrees in general have larger uncertainty bars than the measurements at 0 or 180 degrees.

We have altered the manuscript to include more information on this point and clear up any confusion as to how the phase velocity measurements in these figures are being presented, and to include the necessary caveats that an anisotropic noise distribution increases the uncertainty when trying to analyse the azimuthal variation of phase velocities. These alterations can be found on page 13 line 20 – page 14 line 2.

6. Why did the authors chose to not invert the group velocities for group-velocity maps The bulk of the work is already done by manually measuring all dispersion curves. The methodology would be very similar to the approach based on phase velocities and using the group velocities may provide additional insight or help better constrain the imaged features.he manuscript has been updated with all of this information in order to better explain this process to the reader, and can be found on page 6 lines 16 – 23.

The fast marching method is a wavefront tracking approach that uses an eikonal solver at its core to compute the wavefield through the velocity model at each iteration. As such, it is strictly only valid for tracking phase velocity wavefronts, rather than group velocity. This is due to the fact that the direction of the actual wave vector (phase velocity) and the ray path of the group velocity can be very different in an anisotropic medium (Tanimoto 1987). In practice, one can get away with using an eikonal solver on group velocities if the medium is smooth and only weakly anisotropic. In our case, we judge

this to be a poor assumption in the current study area, due to the complex structure of the large fault system and the clear presence of azimuthal anisotropy in our phase velocity measurements. Group velocity measurements also tend to be associated with larger uncertainties than phase velocities (Lin et al., 2013).

In fact, in an early version of this manuscript submitted to another journal, we did invert group velocity measurements, but we were asked to instead target phase velocities by the reviewers there, for the reasons stated above. As we have received such conflicting advice on this point from several reviewers, we believe that the inversion of group velocities is on shaky ground in this study, and there appears to be no consensus within the community as to whether such a treatment is acceptable. For these reasons, we prefer to focus only on the phase velocities here.

> Tanimoto, 1987. Surface-wave ray tracing equations and Fermat's principle in an anisotropic earth. *Geophys. J. R. Astr. Soc.* 88.

> Lin et al., 2013. Surface wave tomography of the western United States from ambient seismic noise: Rayleigh and Love wave phase velocity maps. *Geophys. J. Int.* 173.

7. The used colormap excels at pronouncing differences in the models (Figures 3, 4, 5, 6, 7, S9, and S10), but can lead to misinterpretation, because it is not perceptually uniform and introduces visual discontinuities where there are no discontinuities in the data. I suggest to use another colormap that is perceptually uniform. One collection can be found here: http://www.fabiocrameri.ch/colourmaps.php. Why non-uniform colormaps can be problematic

We have updated the colour maps of both our phase velocity and S-wave velocity images, using the resource suggested by the reviewer here. As alluded to by the reviewer, the new colour map does a poor job of emphasising the structure at long periods in the phase velocity tomography. As such, we have updated Figs. 3 and 4 to show phase velocity up to 5.0 s period. We discuss the reduction in horizontal resolution, and the horizontal averaging of structure, at longer periods due to the increased wavelength of

the surface waves on page 14, lines 12 – 15.

8. The axis labels on Figures 1a, 1b, 3, 4, 6, and 9 are incomplete/wrong. Especially figure 9 is hard to read and understand like this. Similarly, the colorbar labels on Figures 3 and 4 are missing punctuation marks. The colobar label on Figure 6 is unreadable.

We apologise for the technical issues, but this appears to be an issue with the download of the pdf from the Solid Earth discussions site. It seems to be at least in part a function of browser / operating system. Using Chrome on MacOS, I experience the same issues as the reviewer, but with Firefox on a Linux machine, I do not have issues with the pdf or any of the figures. The raw figure files do not have any of these issues. We will ensure that the final figures are in a suitable format to hopefully avoid these problems, and that the final version of the manuscript works properly.

**Minor points**

- Page 2, Line 19-20 / Figure 1: How high is high topography? A colorbar for topography would be helpful in the figure. This would also help to judge whether there may be possible bias in the measurements caused by station altitudes.

A colourbar for the topography has been added to Fig. 1 as requested.

- Page 2, Line 23: Please give a reference for the observations of "Striations and down dip motion on faults".

The reference to Dogan et al., (2014) has been added to this sentence on page 2 line 23.

- Page 4, Lines 26-33 and Figure 2: The authors mention several possible explanations for the near-0 arrivals that are dominant on the ZZ-component, which indeed is commonly observed. Do these also explain the multiplets of horizontal lines around -10s and +10s in the ZZ-panel or do these require another interpretation? Similar features have been observed in Lehujeur et al., 2018 that have been found to be instrument artefacts caused by the used digitisers.

We would like to thank the reviewer for pointing out this publication to us. We were also puzzled by the potential causes of the arrivals at +/- 10 s, and attributed them to either being body wave reflections contained within the noise, or an artefact of the focal spot. We are happy to also include the reviewers reasoning and this citation in a discussion of the multiplets that can now be found on page 5, lines 7 – 9.

- Page 6, Lines 5-8: The 13.5km inter-station distance threshold makes sense for measurements at 1.5s period. In this study, periods up to 10s are used, which would then lead to all station pairs with less 30km (assuming c=3km/s) inter-station distance at 10s. Instead, station-pairs are excluded based on visual inspection for all periods > 1.5s. Why did the authors choose a different approach for 1.5s and all other periods?

This is a result of the manual picking of phase velocity curves used in this study. It is very easy for us to discard an entire period – velocity map on the basis that none of the data it contains are trustworthy, due to the fact that even the shortest period (1.5 s) does not fulfill the wavelength criteria. At intermediate inter-station distance, the situation arises where some of the short period data may be trustworthy, and we are compelled to try to visually extract the useful data from these period – velocity maps, and must exclude only the longer period data for which the wavelength criteria is still not fulfilled.

We have updated page 6, lines 6 – 11 to better describe this process.

- Page 6, Lines 14-16: This part needs more details to be fully reproducible. I under stand that the theoretically computed phase-velocity dispersion curves (from group-velocity curves) are used to constrain the phase-velocity measurements. How are the computed phase-velocity curves constructed? How are the phase velocities measured, i.e., what part of the waveform is used (e.g., zero-crossings, instantaneous phase, ...)? How exactly does the theoretical phase-velocity curve constrain the measured phase velocities? Are the phase velocities picked manually or automatically?

We hope our response to the reviewers first point regarding the details of the phase

velocity picking also addresses this point.

- Page 6, Line 20-22: Mainly because it is not well described how exactly phase velocities are picked, this part is a bit confusing. Depending on the measurement procedure, it may be going in circles as follows: 1) Measure phase travel-time (if measuring e.g., zero-crossings) 2) Compute phase-velocity estimated from inter-station distance and great circle propagation 3) Convert to travel times. Please clarify this section.

We hope that the updated description of phase velocity calculation helps to clarify this section. The procedure is as follows: Phase velocity curves between stations -> travel time between stations -> phase velocity as a function of position, as is stated in the current text.

- Page 7, Line 29: "A total of 20050...for each node...". Does this mean 20050 times the number of nodes (however many that may be) or 20050 distributed over all nodes?

We calculate 20050 models for each node. So the total sum is 20050 * $number_of_nodes. We have updated the text on page 8, line 15--16 to explicitly state this.$

- Page 7, Line 30: Eq. (2) is referenced before being introduced (in the very next sentence). Maybe change the order of these two sentences.

We have switched these two sentences as suggested by the reviewer.

- Page 9, Lines 7-9: Did the authors try using the regional Karahan et al., (2001) model (see Page 6, Lines 16-17) as the initial model for the linearised inversion? Please elaborate on the benefit of using the initial model obtained from the neighbourhood algorithm. Why does the neighbourhood algorithm not converge to the same solution as the linearised inversion and how different are the final models retrieved from neighbourhood and linearised inversion?

We hope that our response to the reviewers second major point covers this comment as well. We have elaborated on our reasoning for the neighbourhood algorithm. As a side note, "convergence" is not a good description of what the neighbourhood algorithm

attempts to achieve. As we state above, the neighbourhood algorithm seeks to obtain a suite of solutions that would acceptably fit the data, rather than "converging" to a best solution. This is facilitated within the neighbourhood algorithm by an allowance for each step in the inversion to "move to" a newly generated model, even if new model provides a worse fit to the data than the previous model. This prevents the inversion from displaying results that may only be taken from a local minimum in data misfit. As such, it is not necessarily expected that the two inversion procedures will produce to the same solution. In essence, the neighbourhood algorithm defines "the solution" in different way to a linearised approach. This is also compounded by the fact that, as stated above, the linearised inversion has a much finer velocity parameterisation than the neighbourhood algorithm.

- Page 14, Lines 15-19 / Figure 9: This section is a bit hard to follow, mainly because the axis labels in Figure 9 are unreadable.

Again, we apologise for the technical issues. When the axes labels are visible, we believe this section is clear enough.

**Supplementary**

We have updated to text S4 to state that the damping parameters were chosen through subjective judgement.

- Text S10 and Figures S14, S15: There seems to be a slight bias of available interstation-azimuths that can not be explained by the N/S-dominant station distribution alone. The Rayleigh wave distributions (Fig. S14) are dominated by slightly NNE/SSW rays (especially at 6s and 8s), while the Love wave distributions (Fig. S15) are dominated by slightly NNW/SSE-rays (well visible at all periods). Do the authors have an explanation for this difference? Maybe this is a sign for bias introduced during the visual inspection and selection of dispersion curves (main text: page 6, line 9) that could be caused by a potential difference in noise source distribution for Love and Rayleigh waves (See e.g., Riahi et al., 2013). On that note, I suggest to add to Text S10 why

there are generally less paths available for 2.0s than 4.0s, as this is not explained by the (in the main text) wavelength-based exclusion alone and a consequence of the visual inspection, I assume. Do you find dominance of higher modes at shorter periods that lead to these periods being preferentially excluded?

The reviewer is correct on this point, as far as our thinking on this issue goes. We attribute the differences between Rayleigh and Love wave ray paths to differences in the noise source distribution of the respective waves. On the note of their being less paths available at 2 s: this is due to noisier measurements that are made at shorter periods. The extra noise is caused by the increased scattering of the short period waves off small heterogeneities in the upper crust, as well as difficulties in identifying a fundamental mode signal, as the reviewer suggested here.

We have updated Text S11 to include a short discussion of the above points.

**Technical corrections**

- Figure 1: Axis labels for both maps are unreadable. I suggest to add a colorbar for topography. I did not find the station names being used in the text, therefore the authors could remove the explanation.

- Figure 3: Axis labels for all maps are unreadable. Colorbar labels have no punctuation marks and colorbar has no title (phase velocity).

- Figure 4: Same as Figure 3.

- Figure 6: Same as Figure 3  4. Additionally, the colorbar labels are incomplete.

- Figure 9: Same as Figure 3, 4, 6

The errors with the axes labels are likely due to the problems associated with the download from the website on certain browsers and operating systems. We have added the colour bar for the topography as requested. We prefer to keep the description of station names in the manuscript, as the data are in the public domain, and an interested

reader may wish to track it down and use it upon seeing the manuscript.

- Figures S1, S2: Both figures are a bit low quality.

Unfortunately only low resolution .jpg files are available for these figures. We have reduced the size of the figures in an attempt to improve the quality of the images.

- Figure S13: "The blue line is the best fitting curve the raw data." -> "The blue line is the curve best fitting the raw data."

We have updated the figure caption.

---

## Author Comment (AC2) · 6 Dec 2018

Dear Anon,

Thank you for taking the time to review our manuscript. We are glad that you find it interesting and impactful. Below I give our responses to each of the comments on the manuscript. Any page and line references refer to the clean, updated manuscript. I will upload a 'tracked changes' version as a supplement shortly, once the manuscript has been finalised.

Although they emphasize the importance of good and reliable knowledge on crustal

structures along the continental shear deformation zones at the very beginning in the introduction, and since this is one of the primary task for taking all such efforts in the region, I am very upset why they avoid to interpret their results mainly around this target which could be vitally important for future studies that aim at a decent seismic scenario for the region.

There have been numbers of recent geophysical model and observations in a region including the study area and further west dealing with the branch of the NAFZ beneath the Sea of Marmara. However, introduction significantly lacks of a compilation of previous studies and their findings including the DANA experiment.

We appreciate that we may have missed references to studies that would be appropriate to cite and discuss in the context of this work. However, here the reviewer has not given us any specific examples of what essential references might be missing, which makes this comment difficult to respond to.

We believe that we have made extensive referencing to the relevant literature throughout both the introduction and discussion sections of the manuscript. We have made sure to include classic studies of the geological structure of the Izmit region from authors such as: Sengor, Yilmaz, Okay, Barka, Akbayram, Tank, Kahraman, Altuncu Poyraz and Komazawa. We also include a reference for all previous studies using the DANA network, and discuss them where they are relevant. In this revision, we further include a citation to Papaleo et al. (2018), which has been published since the original writing of this manuscript.

In the present work, inversion results for shear wave velocity for deeper sections at 3.5 and 5.5 km do not provide profound velocity contrasts among three tectonic zones, namely, Istanbul, Armutlu-Almacik, and Sakarya Zones (see Fig. 6) whereas using the same network and teleseismic P-and S arrivals Papaleo et al. (2017, 2018) were showing clear separation reflected as relatively high wave speeds beneath Istanbul Zone to the north, and low beneath Sakarya Zone to the south that is mostly likely due

to the lithological differences down to, at least, the depth of 20 km.

Only for the first 1.5 depth range, resolution is sufficient to resolve shear zones along the northern branch. There down to the depth of 1.5 km major difference is claimed by the authors to be associated with low S-wave velocity to the north of the NAFZ, associated with faulted marine clastic sediments near Izmit (Akbayram et al., 2016) and with the Adapazari sedimentary basin. I think a detailed introduction with more geological constraint as well as other geophysical data to support this and further velocity variations at this depth range is missing. Such introduction is crucial since below this range velocity variation does not show high resolution details.

The best horizontal and vertical resolution claimed by the teleseismic tomography of Papaleo et al. (2017, 2018) is 15 km, which greatly exceeds even the maximum depth extent covered by this surface wave study (10 km). Furthermore, as teleseismic tomography studies, Papaleo et al. (2017, 2018) severely lack resolution for near surface structures, notably being unable to detect even the sedimentary basins, due to their lack of crossing ray paths at shallow depths. As such, the depth ranges of the current study and Papaleo et al. (2017, 2018) do not even overlap, and we do not find it surprising that exact comparisons are difficult to draw between the two studies.

We compare the results of our investigation to those of Papaleo et al. (2017, 2018) in the manuscript on page 14 lines 23 – 32, noting that despite the major differences in depth range and resolution between the two studies, some features (such as the high velocity Armutlu Block) are common to both models. We are also open about the fact that our horizontal resolution decreases as a function of depth, due to the increasing period of surface waves used to provide the constraints on the deeper sections of our model. This feature is common to all surface wave tomography studies, and is discussed on page 14, lines 12 – 21. We have also now included full resolution kernels (Fig. S9) in the supplementary material for several nodes in our model at reviewer request, so that this information is available to the reader in a quantitative sense.

It seems there is an effect of N-S elongated azimuth of station pairs on resolved images. This effect can be investigated using sensitivity analysis, i.e., checkerboard test results. I am aware that authors have already added materials in Supplementary but I believe it is much better if given within the Sensitivity Analysis section of the main text. In this way, later they can use this by putting quantitative arguments when they describe the results (reliability of various features which will be potentially examined in the Discussion). I would like to see the ray-paths of periods and their checkerboard results in supplementary file to be able to see the influence of dominance of N-S orientation of stationpairs in your data set.

We appreciate that sensitivity analysis is an important part of appraising the results of a tomographic study. However, we prefer to keep this information in the supplementary material, rather than the main manuscript, and we note that the second reviewer of this paper appears to hold the same opinion.

The revised supplementary material now contains an expanded analysis of horizontal resolution, including spike tests (Fig. S12) and we also demonstrate the recovery of a known random velocity field (Fig. S13). We also include depth resolution kernels in Fig. S9. We hope that these further demonstrations of model resolution will satisfy the reviewers concerns on this point.

According to my recollection, in some studies dealing with ambient noise inversions in the literature, group velocities and related time information are used for further inversion process. Here authors are using phase velocities. Perhaps this has to be addressed in the text.

We refer to our response to the comment by Sven Schippkus regarding group velocity for this point. To restate here: we do not believe that a group velocity tomography is theoretically justified given that we use an eikonal solver, and we have received a wide range of conflicting advice from reviewers of the manuscript on this point.

Figure 2 is interesting. One of the first things that is prominent on this figure is the

zero-offset energy. What might be the major source for that? Needs to be clarified.

We discuss potential sources of the zero offset energy on page 4, line 27 – page 5, line 3.

Azimuthal anisotropy Large scatter azimuthal variations of phase velocities (see Fig. 8 S13) under the presence of N-S dominating azimuth of station-pairs. Thus long period behavior of directional dependent phase velocities is doubtful. And thus, a frequency varying fast velocity directions (with increasing uncertainties as period increases) is also not too convincing.

This work examines anisotropy issue with a superficial discussion regarding early constraints on seismic anisotropy in the region. Authors appear to take the discussion regarding seismic anisotropy only using a single SKS splitting study (Biryol et al., 2010), which has been informative for upper mantle anisotropy. However, there are a few earlier studies performed along the NAFZ (central and western NAFZ) with direct observation of crustal anisotropy. No specific discussion in the light of earlier works revealing upper crustal anisotropic structure mainly based on shear wave splitting structure (e.g. Peng and Ben-Zion, 2004-2005; Hurd and Bohnhoff, 2012) or entire crust from RFs analyses (Vinnink et al., 2015; Licciardi et al. 2018). The question on what part(s) of the area may indicate structure-induced, and what part(s) stress-induced anisotropy is still ambiguous. Moreover, a single model for such a complicated tectonic setting with significant lateral heterogeneities cannot be represented a single-smooth depth-varying model with very consistent SKS orientations (see e.g. Peng and Ben-Zion, 2004-2005; Hurd and Bohnhoff, 2012; Vinnink et al., 2016). At least early shear wave splitting and RFs data suggests the opposite what the current work says.

We thank the reviewer for providing these references to interesting prior studies on the azimuthal anisotropy of the Izmit region. We had not been able to locate them ourselves, and we have incorporated each into the discussion of our anisotropy results in section 4.2, page 16, line 26 – page 17, line 5. However, we do not agree that these

studies suggest the opposite of our results. In fact, Peng and Ben-Zion (2004, 2005) clearly detect a cluster of fast directions oriented between 45 and 90 degrees from north in the top 3 km of the crust. This range of fast directions matches exactly to the Rayleigh wave fast directions that we measure (Fig. 9). Peng and Ben-Zion (2004, 2005) also note that the fast direction often aligns parallel with the strike of the North Anatolian Fault, and we argue the exact same point for our short period measurements on page 17, line 6.

Furthermore, Hurd and Bohnhoff (2012) analyse only one station that overlaps with the current study area: CAY. Their analysis of shear wave splitting at CAY shows fast directions that are aligned between 30 – 90 degrees from north, with most observations clustered at 45 degrees. This again overlaps exactly with our range of measurements in Figs 8 and 9. Whilst Vinnik et al. (2016) targets anisotropy in the upper mantle, they also detect a dominant fast direction of 60 degrees from north between 30 km depth and the surface. We believe these results actually lend great weight to our first order observations of azimuthal anisotropy.

We feel that our description of the anisotropy results may be leading to some confusion here, especially as Fig. 9 was not properly described in the initial submission of this manuscript. We have now updated section 3.6, page 13, lines 2 – 10 to better describe the anisotropy results and to better integrate the information contained in Fig. 9

Another thing I could not figure out is that authors do not provide any clue regarding radial anisotropy? If they are already able to invert both love and Rayleigh wave wouldn't it be possible to visualize radial and tangential shear wave speed variations at various depth?

It is difficult for this study to accurately measure the presence of radial anisotropy as a function of position, due to the differing levels of damping applied to the Love and Rayleigh wave phase velocity tomographies (Fig. S4). The Love wave data require a higher level of damping than the Rayleigh wave data. This differing level of damping

introduces biases between the two velocity data sets that are impossible to resolve from radial anisotropy. In an early version of this manuscript submitted to another journal, we made an attempt to quantify radial anisotropy as the reviewer has requested here. This approach was met with harsh criticism for the reasons outlined above, and as such we do not believe we can make any reliable estimation of radial anisotropy from the results we present in this study.

More importantly, I am seriously wonder why they have not gone for a detailed harmonic analysis that can provide depth variation of fast polarization azimuths on a finer spatial resolution using on available data set.

We do not believe that this data set is suitable for harmonic decomposition, given the already limited ray path distribution (Fig. S17 and S18). Further decomposing the data set is likely to exacerbate the issue with the north-south dominated ray distribution, leading to unreliable estimates of azimuthal anisotropy. We believe that our simpler, first order approach to analyse the data set as a whole is on much safer ground. If the reviewer is unconvinced by our analysis of the broad regional pattern of anisotropy (as indicated by a previous comment above), then we doubt a more detailed regional decomposition would convince them further!

I would omit this part unless it is supported with a more convincing and detailed analysis of the data set.

We strongly believe that our simple analysis of raw phase velocity measurements is a reliable first order measurement of azimuthal anisotropy. This is clearly demonstrated by the fact that our results are in very close agreement with all of the previous shear wave splitting studies that we have been pointed to by this reviewer. As such, we would strongly argue for its inclusion in this manuscript.

Figures For Figs. 1, 3, 4, and 6, values of latitude and longitude is strange.

This may be due to the fact that Fig. 1 actually displays a larger geographical area

than the subsequent figures. We have checked carefully, and the latitude and longitude values on each figure are correct.

Two references of Sengor ( Sengor and Yilmaz, 1981; ÂÿSengor et al., 2005) are not listed in the alphabetical order.

This is probably due to the bibliography style file not recognising the Turkish "S" character. This issue has been fixed.

---

## Author Comment (AC4) · 10 Dec 2018

**Introduction**

This supporting information provides figures related to the choice of pre-processing regime applied to our noise recording prior to the calculation of the cross correlation functions (Figs. S1 and S2). We show examples of the group velocity - period diagrams used when creating the phase velocity data set (Fig. S3). We also include a figure illustrating our choice of damping parameter in the Rayleigh and Love wave phase velocity inversions (Fig. S4). Fig. S5 and S6 show the data misfit information for our phase velocity tomography. Figs. S8 - S15 provide information on the vertical and horizontal resolution of of our phase velocity tomography through sensitivity kernels and checkerboard tests. Fig. S16 shows the azimuthal variation of our Love wave phase velocities.

**Text S1. Effect of noise pre-processing on the signal-to-noise ratio of the cross-correlation functions**

Fig. S1 shows the evolution of the signal-to-noise ratio of the cross correlation functions as a function of the amount of data stacked. The signal-to-noise ratio is defined as the ratio between the amplitude of the surface wave arrival, and a 20 s long window of coda energy that arrives late in the correlation function. We show examples for four different noise pre-processing schemes. We test the effects of the noise window duration (1-hour or 4-hours long), as well as different forms of amplitude normalization (1-bit normalization or amplitude clipping). It is clear that the choice of noise pre-processing has little effect on the final signal-to-noise ratios. All pre-processing approaches result in an average signal-to-noise ratio of $\sim 16$ once all data have been stacked. This analysis was performed on a subset of the data that spanned a period of 3-months. The final tomography utilized data from the full 16-month long deployment of the DANA network.

**Text S2. Effect of noise pre-processing on the coherency of the cross-correlation functions.**

Fig. S2 shows the power spectral densities of the cross correlation functions under the four different processing schemes (Text S1.). As with signal-to-noise ratio, there is little variation in coherency between the different approaches. However, amplitude clipping does appear to have a slightly higher power spectral density than the 1-bit normalization approach. The 4-hour long windows are also slightly more coherent than the 1-hour long windows. Given this, we choose the pre-processing scheme that implements amplitude clipping on noise windows that are 4-hours long.

**Text S3. Examples of group velocity-period diagrams.**

Fig. S3 shows four examples of group velocity - period diagrams that were used in the study. We show examples of Rayleigh wave diagrams taken from the radial-radial cross-correlation functions between stations with inter-station distances of 17 km and 61 km. We also show transverse-transverse cross-correlation functions taken for stations separated by 22 km and 65 km. On the 17 km distance RR component, group and phase velocity values are not picked for periods longer than 4.0 s, as the measurements at long periods are deemed to be unreliable due to the wavelength criterion. On the 22 km TT component, velocity values are not picked above 5.0 s period for the same reason. The group velocity values begin to increase rapidly up to unrealistic values for periods greater than these cut-offs.

**Text S4. Effect of damping on the phase velocity inversion.**

Fig. S4 shows model variance vs. RMS travel time residual as a function of damping parameter for the final models produced by the Rayleigh and Love wave phase velocity inversions at 4.0 s period. We select, through subjective judgement, the level of damping that gives us a substantial decrease in model variance for only a modest increase in data misfit. We trialled damping parameters with a range between $\epsilon = 0$ and $\epsilon = 500$.

[Figure]

**Figure S1.** Effect of noise pre-processing on the signal-to-noise ratio of the cross correlation functions as a function of the amount of data stacked. Each panel shows a different pre-processing scheme. Top left: 4-hour long noise windows with amplitude clipping. Top right: 4-hour long noise windows with 1-bit normalization. Bottom left: 1-hour long noise window with amplitude clipping. Bottom right: 1-hour long noise window with 1-bit normalization. Grey lines represent individual cross correlation functions, and the solid black line represents the mean of all correlation functions. The dashed blacked line indicates one standard deviation.

**Text S5. Data misfit to the Rayleigh and Love wave phase velocity tomography.**

Figs. S5 and S6 show the data misfit of the phase velocity tomography for Rayleigh and Love waves, respectively. We show the misfit of the initial, constant velocity model prior to the tomographic inversion, and the data misfit of the final phase velocity model at 2.0, 4.0, 6.0 and 8.0 s period. The final phase velocity models are shown in Fig. 3 and 4 in the main text.

5    **Text S6. Data misfit of the S-wave velocity inversion.**

Fig. S7 shows the fit of the individual phase velocity dispersion curves at three nodes in the S-wave velocity model (Fig. 5 in the main text). We show the dispersion curves that are extracted from the phase velocity tomographies for both Rayleigh and Love waves, and the corresponding dispersion curves predicted by our final S-wave velocity model. The improvement of fit provided by the linearised inversion over the neighbourhood algorithm result is shown in each row.

[Figure]

**Figure S2.** Effect of noise pre-processing on the coherency of the cross correlation functions. Each panel shows a different pre-processing scheme. Top left: 4-hour long noise windows with amplitude clipping. Top right: 4-hour long noise windows with 1-bit normalization. Bottom left: 1-hour long noise window with amplitude clipping. Bottom right: 1-hour long noise window with 1-bit normalization. Grey lines represent individual cross correlation functions, and the solid black line represents the mean of all correlation functions.

**Text S7. Rayleigh and Love wave sensitivity kernels.**

Fig. S8 shows the partial derivatives of phase velocity with respect to S-wave velocity as a function of depth at each grid point is our final isotropic S-wave velocity model. We show period of 2.0 s, 5.0 s and 8.0 s. This illustrates the depth sensitivity of our phase velocity observations and constrains the depth resolution of our S-wave velocity model. Our long period Rayleigh
5   wave observations maintain sensitivity up to 10.0 km depth, whilst our Love wave observations are sensitive to more shallow structure, mostly above 5.0 km depth.

**Text S8. Depth resolution of the S-wave velocity model**

Fig. S9 shows the final resolution kernels of the S-wave velocity model, after the linearised inversion. It is clear that the structure closest to the surface is best resolved by the data set, which is to be expected for surface wave studies, where resolution at depth
10   depends on observations from longer period waves. This effect is most obvious within the Istanbul zone, where the pervasive low S-wave velocities near the surface caused by the presence of sedimentary basins results in relatively lower resolution below $\sim 3$ km depth.

[Figure]

**Figure S3.** Examples of period - group velocity maps for four inter-station pairs. The inset dispersion curves are the corresponding phase velocity curves that were picked, shown in red dots. Red colours indicate a larger amplitude for the envelope of the wave at the given period, for each velocity. The black squares indicate the likely group velocity values, which are picked manually. We show dispersion maps for both short and long distances (< 25 km and > 60 km) for both Rayleigh (left column) and Love (right column) waves. The bars of the right hand side of each period - group velocity map shows the cross correlation function.

[Figure]

**Figure S4.** Top: Effect of damping on the Rayleigh wave phase velocity inversion at 4.0 s period. Bottom: Effect of damping on the Love wave phase velocity inversion. Each black dot represents a trial of a different damping parameter between 0 and 500. The red lines indicate the model variance and data misfit of the chosen damping factor for the inversion, which is indicated on each figure.

**Text S9. Checkerboard tests for the Rayleigh and Love wave phase velocity.**

Fig. S10 shows checkerboard tests that illustrate the horizontal resolution of the Rayleigh wave phase velocity tomography between 3.0 and 9.0 s, Fig. S11 shows the same image for the Love wave tomography. The maps were obtained by inverting for the original checkerboard pattern from a constant velocity starting model using a noisy synthetic data set with the same
5  ray path distribution as our tomography. The standard deviation of the white noise added to the input travel times is 1.5 s. We maintain good horizontal resolution at all periods, though there is some smearing at long periods (9.0 s) for both Rayleigh and Love waves. In general, the checkerboard pattern is recovered better using the ray distribution of the Love wave tomography, rather than the Rayleigh wave tomography. The corresponding ray path distributions are shown in Figs. S14 and S15. In order to test the robustness of our phase velocity inversion against anomalies of different shapes and wavelengths, we also include
10  tests to recover a velocity model consisting of spikes (Fig. S12) and a randomly generated velocity model (Fig. S13). Both of these tests were carried out on the Rayleigh wave ray path distribution at 4 s period.

[Figure]

**Figure S5.** Data misfit of the Rayleigh wave phase velocity tomography at periods 2.0, 4.0, 6.0 and 8.0 seconds. We show the travel time residual in 0.25 s bins, and the number of measurements that fall in each bin. The left hand column shows the misfit of the original, constant velocity inversion, and the right hand column shows the misfit of the final tomography model. Each histogram is labelled with the corresponding root mean square misfit and variance of the travel time residuals.

[Figure]

**Figure S6.** Data misfit of the Love wave phase velocity tomography at periods 2.0, 4.0, 6.0 and 8.0 seconds. We show the travel time residual in 0.25 s bins, and the number of measurements that fall in each bin. The left hand column shows the misfit of the original, constant velocity inversion, and the right hand column shows the misfit of the final tomography model. Each histogram is labelled with the corresponding root mean square misfit and variance of the travel time residuals.

[Figure]

**Figure S7.** Dispersion misfit of the S-wave velocity inversion at the three nodes shown in Fig. 5 (main text). The fit for Rayleigh wave phase velocities is shown in the left column, Love wave fits are shown on the right. The target dispersion curves from the phase velocity inversion is shown in black, and the red curves show the dispersion calculated from our final S-wave velocity model. The improvement in data fit at each node that results from applying the linearised inversion to the result of the neighbourhood algorithm is indicated on each row, including the final average data residual.

[Figure]

**Figure S8.** Top: Partial derivatives of Rayleigh wave phase velocity with respect to S-wave velocity at each grid point in our final S-wave velocity model. Bottom: Partial derivatives of Love wave phase velocity with respect to S-wave velocity at each grid point in our final S-wave velocity model. Black lines show sensitivity at 2.0 s period, blue lines 5.0 s period and red lines 8.0 s period.

[Figure]

**Figure S9.** Resolution matrix of the final, linearised S-wave velocity inversion. Red colours indicate high resolution, and blue colours indicate lower resolution. The ideal resolution matrix would consist of the identity matrix - a perfectly diagonal matrix. The further from identity the resolution matrix becomes, the larger the trade offs between different depths in the model.

**Text S10. Azimuthal anisotropy of Love wave phase velocities.**

Fig. S16 shows the azimuthal variation of the Love wave phase velocities between 0 and 180 degrees from north. The measurements have a higher variance than the Rayleigh wave counterparts, and the dominant fast direction varies between 25 and 40 degrees for the $2\theta$ component. The $4\theta$ component generally has a fast direction between 85 and 120 degrees. The average amplitude of the $2\theta$ component is 0.036 km s$^{-1}$ 1.3%), whilst the $4\theta$ component has an average of 0.025 km s$^{-1}$ (0.9%).

**Text S11. Rose diagrams of Rayleigh and Love wave propagation azimuths.**

Figs. S17 and S18 shows the azimuthal distribution of ray coverage for both Rayleigh and Love waves. The azimuthal distribution of ray coverage shows a strong bias to north-south oriented paths in both the Rayleigh and Love waves. The azimuthal distribution could be indicative of the dominant direction of noise propagation at the DANA network (oriented towards the Black Sea and Mediterranean Sea), but is most likely the result of the rectangular array shape, and subsequent better sampling of north-south paths after short paths have been excluded. There is evidence of slight heterogeneity between the Rayleigh and Love wave azimuthal distributions. Rayleigh waves show a slight bias towards NNW - SSW oriented paths, whilst Love waves are more clustered along NNE - SSE ray paths. This could be indicative of a difference in noise source distribution for these two wave types. The fact that less measurements are available at 2.0 s period than at higher periods can be explained by the generally noisier measurements made at short period, for reasons such as higher structural heterogeneity, and the appearance of higher mode surface waves.

**Text S11. Description of the final S-wave velocity model file.**

The final S-wave velocity model is included as a separate file. This file is an ASCII plain text document and is organized with each row representing: Longitude, Latitude, Depth, S-wave velocity. Each row represents a different node in our final model.

**Rayleigh checkerboard tests**

[Figure]

**Figure S10.** Checkerboard tests for the Rayleigh wave phase velocity tomography between 3.0 s and 9.0 s. The top left panel shows the original checkerboard that we attempt to retrieve using a noisy (Text S9.) synthetic data set with our ray path distribution at each period. Thick black lines represent the locations of the mapped faults. The thin black line represents the Sakarya River.

**Love checkerboard tests**

[Figure]

**Figure S11.** Checkerboard tests for the Love wave phase velocity tomography between 3.0 s and 9.0 s. The top left panel shows the original checkerboard that we attempt to retrieve using a noisy (Text S8.) synthetic data set with our ray path distribution at each period. Thick black lines represent the locations of the mapped faults. The thin black line represents the Sakarya River.

[Figure]

**Figure S12.** Spike test for the Rayleigh wave velocity at 4.0 s period. The left panel shows the original spike pattern that we attempt to retrieve using a noisy (Text S8.) synthetic data set with our ray path distribution. Thick black lines represent the locations of the mapped faults. The thin black line represents the Sakarya River.

[Figure]

**Figure S13.** Random model test for the Rayleigh wave velocity at 4.0 s period. The left panel shows the original random pattern that we attempt to retrieve using a noisy (Text S8.) synthetic data set with our ray path distribution. Thick black lines represent the locations of the mapped faults. The thin black line represents the Sakarya River.

**Rayleigh**

**2.0 s**

**4.0 s**

**6.0 s**

**8.0 s**

[Figure]

**Figure S14.** Ray path distribution for the Rayleigh wave phase velocity tomography between 2.0 s and 8.0 s. Red triangles show stations of the DANA network. The black lines connecting them are the ray paths. Thick black lines represent the locations of the mapped faults. The thin black line represents the Sakarya River.

**Love**

[Figure]

**Figure S15.** Ray path distribution for the Love wave phase velocity tomography between 2.0 s and 8.0 s. Red triangles show stations of the DANA network. The black lines connecting them are the ray paths. Thick black lines represent the locations of the mapped faults. The thin black line represents the Sakarya River.

[Figure]

**Figure S16.** Azimuthal variation of Love wave phase velocities with propagation azimuth (from north). Black dots indicate the raw phase velocity measurements, large red dots show the average of the phase velocities within 5 degree azimuth bins, and the corresponding standard error of the mean for the bin. The blue line is the curve (eq. 3) that best fits the raw data (black dots). $u_0$ is the average (isotropic) phase velocity. We show the root mean square misfit of the blue curve to the phase velocity measurements, as well as the variance of the residuals. We indicate the $2\theta$ and $4\theta$ amplitudes and fast directions that correspond to the blue curve. The azimuthal distribution of ray paths used in this analysis is shown i supplementary Fig. S15.

[Figure]

**Figure S17.** Azimuthal distribution of Rayleigh wave ray coverage at 2 s, 4 s, 6 s and 8 s. The azimuth from north is indicated on the outside of the rose diagram. The data are split into 10 degree azimuth bins. The x-axis indicates the number of rays in the given azimuth bin.

[Figure]

**Figure S18.** Azimuthal distribution of Love wave ray coverage at 2 s, 4 s, 6 s and 8 s. The azimuth from north is indicated on the outside of the rose diagram. The data are split into 10 degree azimuth bins. The x-axis indicates the number of rays in the given azimuth bin.

---

## Author Response (AR2)

Dear Irene Bianchi,

Thank you for your suggestions as to how to improve the manuscript. We have given each comment due consideration, and our responses are listed below, along with our changes to the manuscript as a result of your feedback. In addition, we have made alterations to the text to improve the readability of the manuscript. Any page and line numbers refer to the marked-up 'track changes' pdf file.

The editor's suggestions pertain to a previous comment made by an anonymous reviewer, which we will quote where appropriate, along with our revised responses. In bold text, we highlight a few key excerpts from the updated manuscript that we feel most directly address the editorial and reviewer comments.

Editorial comment: "1) some more introduction in why it is important to give constraints on anisotropy in the study area,"

Reviewer comment: "…there are a few earlier studies performed along the NAFZ (central and western NAFZ) with direct observation of crustal anisotropy. No specific discussion in the light of earlier works revealing upper crustal anisotropic structure mainly based on shear wave splitting structure (e.g. Peng and Ben-Zion, 2004-2005; Hurd and Bohnhoff, 2012) or entire crust from RFs analyses (Vinnink et al., 2015; Licciardi et al. 2018)."

We have updated and expanded the introduction to address this point. Specifically, we outline the importance of providing the constraints on anisotropy along the North Anatolian Fault on page 4, beginning on line 1:

"**Observations of azimuthal anisotropy in the upper crust can provide insights into the state of tectonic stress within a region, and potentially the orientation of pervasive mineral fabric and the structural influence of major faults (e.g. Hurd and Bohnhoff (2012), Polat et al. (2012)). Such information provided by azimuthal anisotropy is particularly important in areas such as the North Anatolian Fault, where in-situ stress observations are rare, and extensive deformation occurs off of mapped faults (Bouchon and Karabulut, 2008; Altuncu Poyraz et al., 2015)**."

We also include both new introduction (page 4) and an extended discussion (page 19 - 20) of the several studies suggested by the reviewer (Peng and Ben-Zion, Hurd and Bohnhoff, Licciardi et al. and Vinnik et al.). Our introduction and discussion now includes a presentation of the results provided by these relevant studies:

"…**Peng and Ben-Zion (2004) and Peng and Ben-Zion (2005) also display a seismic fast direction in the upper crust that clusters between 45◦ and 90◦ from north, often aligning parallel to the strike of the North Anatolian Fault.** "

"…**Hurd and Bohnhoff (2012) at the station CAY, located within our study region to the east of Lake Sapanca (Fig. 1), also showed directions between 30◦ and 90◦, with the majority falling between 40◦ and 50◦. Further east, the fast polarisation directions measured by Hurd and Bohnhoff (2012) are more commonly aligned NW – SE.**"

Editorial comment: "2) how the detection and characterization of anisotropy might help addressing some major unresolved issue about this area, also related to seismic hazard"

We have expanded the introduction to describe the major unresolved issues relating to anisotropy for our study region. On page 4, lines 7 – 10, we specifically outline the mechanisms which may be expected to cause seismic anisotropy in the area (stress vs. structure control). We also introduce to findings of previous studies such as Peng and Ben-Zion (2004), Hurd and Bohnhoff (2012), which suggest an unresolved discrepancy between the observed fast directions and the direction of maximum compressive stress for the Izmit-Adapazari region (page 4, lines 8 – 16):

"**Earthquake focal mechanisms suggest that the direction of maximum compressive stress in the Izmit-Adapazari region is oriented NW – SE, between 120° – 160° from north (Bohnhoff et al., 2006). If the regional anisotropy is primarily stress-controlled, we would expect the seismic fast direction to be aligned in the direction of maximum compressive stress, due to the preferential closure of fractures in this direction (Crampin and Lovell, 1991). However, Peng and Ben-Zion (2004) used local seismicity to show that the fast polarisation direction at stations close to the ruptured Düzce fault (Fig. 1) are generally parallel to and vary with the fault strike, suggesting an anisotropy mechanism determined by deformation fabric.**"

We have made clear the link between seismic anisotropy and deformation fabrics that form within the upper crust, which directly relates to the regional tectonics and seismic hazard (page 4).

We have expanded our discussion section to include the relation of our findings to the previous studies of Peng and Ben-Zion, Hurd and Bohnhoff and Licciardi et al., and the new information provided by our observations (pages 19 - 20).

"**A dominant fast direction between 50° – 90° (NE – SW) from north (Fig. 9) indicates that the anisotropy in the region is likely structure-controlled. This observation was also noted in anisotropic receiver functions by Licciardi et al. (2018), who found that the fast shear wave polarisation directions along the central portion of the North Anatolian Fault align with the strike of mapped faults at stations located close to those faults, implying structure-controlled anisotropy.**"

Editorial comment: "3) is the anisotropy stress- or structure-controlled? Can you give some inferences?"

Reviewer comment: The question on what part(s) of the area may indicate structure-induced, and what part(s) stress-induced anisotropy is still ambiguous… At least early shear wave splitting and RFs data suggests the opposite what the current work says.

We feel that our results strongly suggest a structure-controlled anisotropy in the vicinity of the North Anatolian Fault. We have included this conclusion in our discussion section (page 19). In this expanded section, we also compare our findings to those of previous shear wave splitting studies (Peng and Ben-Zion (2004, 2005), Licciardi et al. 2018) and note that, in our observation and those of previous studies, stations close to the fault strand appear to show structure-controlled anisotropy, whereas stations more distant to the fault more commonly display a fast direction parallel to the direction of maximum compressive stress (page 4, lines 10 – 19). We believe that the reason for the variance in the measurements of anisotropy made in this region, and the discrepancy between structure and stress-control, are likely to be the

result of whether the measurements are taken within close proximity of the fault (page 19, line 15).

**"Fig. 9 shows a nearly 90◦ fast direction at 2 – 3 s period (~ 0 – 3 km depth) that aligns approximately with the strike of the North Anatolian Fault through the region. This observation clearly implies structure-controlled anisotropy that is dominated by faulting in the very upper crust, similar to the observations of Licciardi et al. (2018) for the top 15 km of the central section of the North Anatolian Fault."**

*For replying to these questions you might refer to some papers like Hurd and Bonhoff 2012; Licciardi et al; 2018, Polat et al, 2012.*

We have given these studies more thorough treatment in both our expanded introduction (page 4) and discussion (page 19) sections.

*Moreover, you might consider to include one more picture in which your results are directly compared to previous results.*

We gave due consideration to the inclusion of a new figure or table with this information. However, the results of the previous studies show a high variability depending upon the location of the station used. In most cases there is not a robust 'average' observation that can be easily displayed in a figure for direct comparison, and often a numerical result is not directly stated by the authors. In general, the construction of such a figure would consist of estimating the previous results visually from the figures, which does not provide a robust result for comparison. Nonetheless, we reference and discuss all the relevant studies for this area of the North Anatolian Fault Zone both in the introduction and discussion of our manuscript (page 4, lines 7, 11, 15, page 18, line 23, page 19 line 31).

We believe that this is the fairest and clearest representation of this information, and that the comparison between our results and the previous studies is made sufficiently clear to the reader.

Thank you for taking the time to consider the manuscript.

Yours sincerely,
George Taylor

[revised manuscript text omitted]